# SAM-Veteran: An MLLM-Based Human-like SAM Agent for Reasoning Segmentation

**Tianyuan Du**[*], **Haopeng Li**[*], **Zhen Fan, Jiarui Zhang, Panwang Pan, Yang Zhang**[†]
PICO
{dutianyuan, lihaopeng.hop, fanzhen.0315, zhangjiarui.zjr123,
panpanwang, zhangyang.0621}@bytedance.com

## Abstract

Significant progress has been made in reasoning segmentation by combining multi-modal large language models (MLLMs) with the Segment Anything Model (SAM): the former excel in reasoning and vision–language alignment, while the latter offers powerful pixel-level understanding. However, current paradigms fall short in exploiting SAM's strengths, especially the ability to support iterative mask refinement by interactive segmentation, a process that human users can naturally perform. To bridge this gap, we introduce **SAM-Veteran**, an experienced mask-aware SAM agent capable of emulating human interaction with SAM via a reasoning-driven segmentation workflow that integrates (i) generating bounding boxes given image–query pairs for SAM input, (ii) proposing refinement points based on SAM-generated masks, and (iii) adaptively terminating the process. Aiming for this goal, we propose a multi-task reinforcement learning framework based on Group Relative Policy Optimization (GRPO), which enhances the MLLM's abilities in textual grounding and mask comprehension. Furthermore, we introduce a dynamic sampling strategy tailored for generating both boxes and points to stabilize training. Extensive experiments across diverse datasets show that SAM-Veteran achieves human-like interaction with SAM and establishes new state-of-the-art performance on both in-domain and out-of-domain benchmarks.

## 1 Introduction

Reasoning segmentation generates pixel-level binary masks by interpreting textual queries through logical reasoning (Yu et al., 2018; Yang et al., 2022; Wang et al., 2022; Liu et al., 2023b; Zou et al., 2023a;b; Yang et al., 2023; Lai et al., 2024; Ren et al., 2024a;b; Rasheed et al., 2024; Liu et al., 2025a). Unlike traditional semantic or instance segmentation, which depends on predefined categorical labels (e.g., *dog* or *baby*), reasoning segmentation is designed to handle more complex, context-dependent queries such as *the object held by the person in red*. This setting better reflects real-world needs for intelligent assistants and robots, while also imposing higher demands on the model. It requires advanced capabilities such as nuanced textual understanding and logical reasoning, fine-grained visual perception and image–text alignment, and broad domain knowledge combined with common sense. Multi-modal large language models (MLLMs) (Li et al., 2023; Liu et al., 2023a; Wang et al., 2024; Chen et al., 2024b; Liu et al., 2024; Yang et al., 2025; Bai et al., 2025), owing to their integrated reasoning and multi-modal perception abilities, have thus become a central component in the latest paradigms for this task.

Recent studies have investigated two primary MLLM-based paradigms for this task: (1) Supervised Fine-Tuning (SFT), where MLLMs generate special tokens that control a learnable segmentation head or decoder, thereby enabling end-to-end training as a unified model (Yan et al., 2024; Lai et al., 2024; Yan et al., 2025); and (2) Reinforcement Learning (RL), where MLLMs are optimized with reward signals for generating boxes and/or points that are then fed into Segment Anything Model (SAM) (Kirillov et al., 2023) to produce the final segmentation (Liu et al., 2025b; Huang et al., 2025). While SFT-based methods effectively incorporate the reasoning capability of MLLMs into

---

[*]Equal contribution.
[†]Corresponding Author.

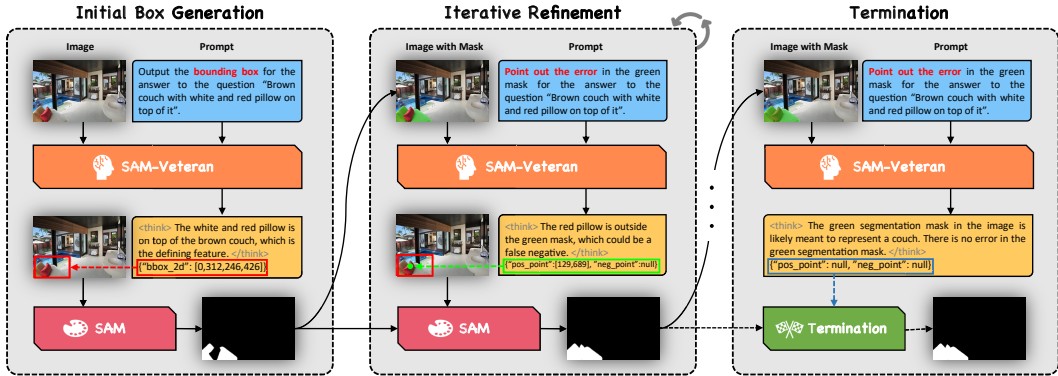

Figure 1: Inference workflow of SAM-Veteran. Given an image-question pair, SAM-Veteran first predicts a bounding box for the target, which SAM converts into an initial mask. SAM-Veteran iteratively refines this mask by generating refinement points for SAM, until either both points are `null` (indicating a satisfactory segmentation) or the maximum refinement step is reached.

segmentation pipelines, they suffer from two key limitations: (i) catastrophic forgetting of general reasoning abilities, and (ii) poor generalization to out-of-domain data (Chu et al., 2025). RL-based methods, on the other hand, either decouple MLLMs from SAM during training, leading to sub-optimal segmentation inputs (Liu et al., 2025a), or fail to fully exploit SAM's capacity for iterative refinement and interactive segmentation as human users (Huang et al., 2025). SegAgent (Zhu et al., 2025b) is the first, and so far the only, approach that emulates the behavior of human SAM users, performing SAM-friendly point generation and iterative mask refinement through RL-style optimization. However, it depends on bounding boxes generated by another MLLM as the initial input for SAM and lacks the ability to adaptively terminate the refinement process as humans do. Moreover, the reliance on hand-crafted ground truth for point traces may constrain its exploration of more effective strategies for mask refinement.

To address these limitations, we propose **SAM-Veteran**, an experienced MLLM-based SAM agent designed for human-like SAM usage. As illustrated in Figure 1, SAM-Veteran follows a complete reasoning-driven segmentation workflow that includes: (1) generating bounding boxes given image–query pairs for SAM input, (2) proposing refinement points based on SAM-generated masks, and (3) adaptively terminating the process. This design closely mirrors the intuitive way humans interact with SAM. To enable the MLLM to perform this workflow, we propose a multi-task reinforcement learning framework based on Group Relative Policy Optimization (GRPO) (Shao et al., 2024), designed to enhance its capabilities in textual grounding and mask comprehension. The framework consists of: (i) a **textual grounding task**, where the model learns to generate bounding boxes that maximize both box IoU and mask IoU of SAM-predicted masks; (ii) a **mask comprehension task**, where the model judges whether the SAM-predicted mask is satisfactory and, if not, generates refinement points to effectively improve mask IoU; and (iii) an **auxiliary task**, which further strengthens mask comprehension by requiring the model to identify the centers of flaws in manually corrupted masks with random false positives and/or false negatives. Moreover, to stabilize GRPO-based reinforcement learning, we incorporate the concept of dynamic sampling from DAPO (Yu et al., 2025) into our framework. Extensive experiments demonstrate that SAM-Veteran effectively generates bounding boxes for initial SAM segmentation, produces points for iterative mask refinement, and can adaptively terminate the process. Evaluated on multiple reasoning-based segmentation datasets, it achieves state-of-the-art (SOTA) performance on both in-domain and out-of-domain benchmarks. The contributions of this work are summarized as follows:

- We propose SAM-Veteran, an experienced SAM agent that emulates human usage behavior through a reasoning-driven segmentation workflow encompassing box generation, iterative mask refinement, and adaptive termination. To the best of our knowledge, it is the first model to unify all of these behaviors within a single framework.

- We present a multi-task RL framework consisting of three tasks: a grounding task, which trains the model to generate bounding boxes well-suited for SAM segmentation; a mask comprehen-

sion task, which enables the model to evaluate mask quality and provide refinement points when necessary; and an auxiliary task, which further enhances the model's mask comprehension ability.

- We conduct extensive experiments on both in-domain and out-of-domain datasets. The results demonstrate that SAM-Veteran performs human-like SAM usage and achieves state-of-the-art performance across multiple benchmarks.

## 2 RELATED WORK

**Supervised Fine-Tuning Approaches.** SFT has been widely adopted to adapt MLLMs for reasoning segmentation (Wei et al., 2024; Zhang et al., 2024a; Bai et al., 2024; Sun et al., 2024; Zhang et al., 2024b; Yuan et al., 2024; Yan et al., 2025). Early efforts, such as LISA (Lai et al., 2024) and its extensions GSVA (Xia et al., 2024), PixelLM (Ren et al., 2024b), and PerceptionGPT (Pi et al., 2024), employ SFT on mixed datasets containing mask labels to finetune large language models (LLMs) utilizing LoRA (Hu et al., 2022) and segmentation modules jointly. Diverging from this paradigm, LLM-Seg (Wang & Ke, 2024) and HReasonSeg (Lin et al., 2025) reformulate the segmentation task as a mask selection problem without the need to train the segmentation modules. These methods rely on implicit semantic tokens or task-specific decoders to generate pixel-level masks, achieving strong performance on in-domain benchmarks. MMCPF (Tang et al., 2024) and GenSAM (Hu et al., 2024) focus on camouflaged object segmentation with SAM by optimizing visual prompts, but neither realizes iterative refinement with mask comprehension.

**Reinforcement Learning Approaches.** RL has emerged as a promising alternative, enabling models to learn generalizable segmentation policies through reward signals (Wang et al., 2025c; Wu et al., 2025; Wang et al., 2025b). Seg-Zero (Liu et al., 2025a) initializes its reasoning model with a pretrained MLLM and employs RL to activate reasoning chains without requiring SFT on reasoning data. VisionReasoner (Liu et al., 2025b) broadens this scope by addressing multiple visual perception tasks, including detection, segmentation, and counting, within a unified GRPO framework. More recently, SAM-R1 (Huang et al., 2025) advances this line of research by directly integrating SAM into the RL feedback loop, where segmentation masks serve as fine-grained reward signals. Beyond purely RL-based methods with frozen segmentation modules, recent work explores hybrid strategies that combine SFT and RL. For instance, POPEN (Zhu et al., 2025a) incorporates preference-based optimization to align MLLMs with human preferences via RL. Besides, SegAgent first generates synthetic trajectories using a modified RL based on StaR (Zelikman et al., 2022), and then finetunes the MLLMs on these trajectories to mimic human interactions with SAM.

## 3 SAM-VETERAN

### 3.1 TASK FORMULATION

In this work, we empower the MLLM to mimic human behavior when using SAM for reasoning segmentation. The optimized MLLM is treated as an experienced SAM user, referred to as SAM-Veteran, which can carry out textual grounding (generate bounding boxes), iterative mask refinement (output point coordinates), and adaptive termination. Specifically, given an image $I$ and a question $Q$, the MLLM is prompted with $Q^{\mathrm{B}}$ to generate a bounding box $b \in \mathbb{R}^4$ for the target object. This image and bounding box are then input to SAM, producing an initial segmentation mask $M$. Next, the image and the resulting mask are fed back into the MLLM, which is now prompted with $Q^{\mathrm{P}}$ to generate refinement points: a positive point $p^+ \in \mathbb{R}^2$ to recover the false negative region and/or a negative point $p^- \in \mathbb{R}^2$ to suppress the false positive region. These points, along with the previous mask, are passed to SAM to generate an updated segmentation $M'$. This iterative process continues until either the MLLM determines that the mask is satisfactory, at which it outputs `null` for both points, or the maximum refinement step is reached (forced termination). This process can be framed as a Markov Decision Process (MDP) $(\mathcal{S}, \mathcal{A}, T, R)$, where:

- **State** $s \in \mathcal{S}$: Represents the current segmentation result. Since the quality of the mask depends on the image and the question, we define the state as a triplet, i.e., $s = (M, I, Q)$.
- **Action** $a \in \mathcal{A}$: Refers to the answer from the MLLM for SAM input. Specifically, the initial action is the bounding box for question grounding, i.e., $a = b$, while the subsequent actions

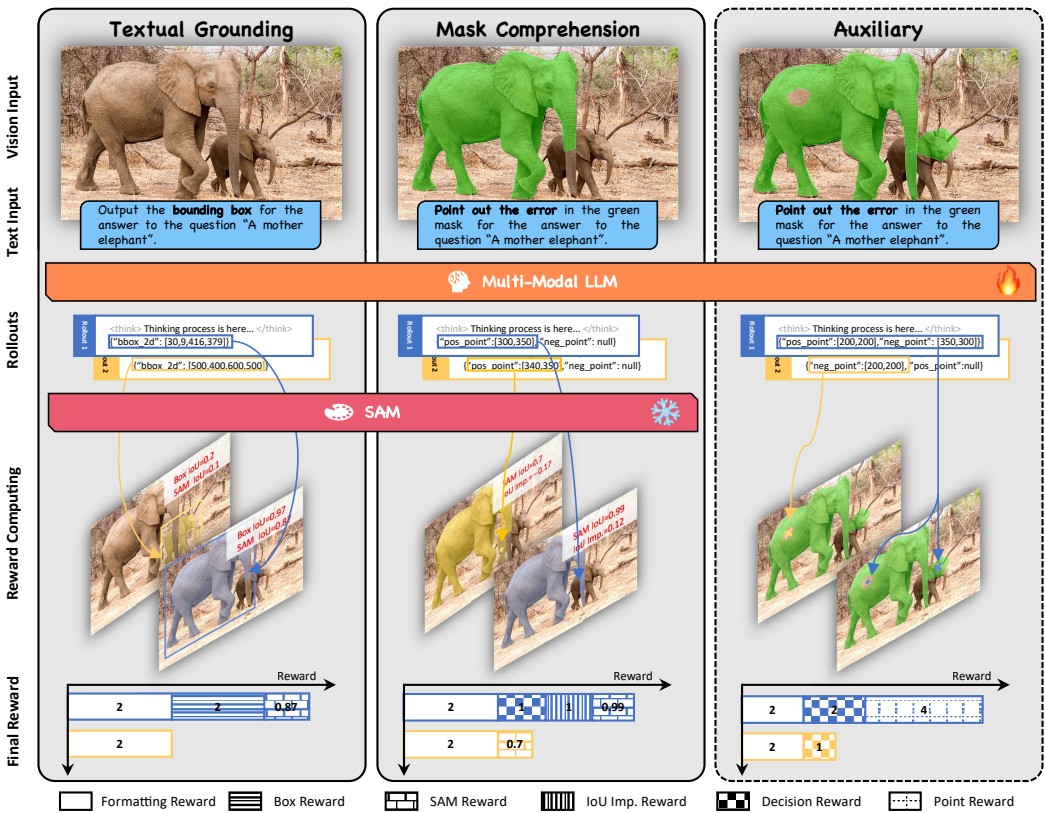

Figure 2: Multi-task RL framework comprising Textual Grounding, Mask Comprehension, and Auxiliary Mask Comprehension. Two rollouts (with their rewards) are shown in different colors (blue and yellow). In the final reward, different bar textures represent different reward functions.

are the points for mask refinement, i.e., $a \in \{(p^+, p^-), (p^+, \texttt{null}), (\texttt{null}, p^-)\}$. When mask refinement is complete, the action is $a = (\texttt{null}, \texttt{null})$.

- **Transition Function** $T : \mathcal{S} \times \mathcal{A} \to \mathcal{S}$: Defines how the current state transitions to the next state given a specific action, i.e., $s' = T(s, a)$. In our approach, the transition function is based on SAM, which generates a new segmentation mask $M'$ given the image $I$, current mask $M$, and current action $a$ from the MLLM.

- **Reward Function** $R : \mathcal{S} \times \mathcal{S} \times \mathcal{A} \to \mathbb{R}$: Comprehensively considering the quality of current/next state and the action, and providing the reward for the current transition, i.e., $r = R(s', s, a)$. The details of the reward function are introduced in the following section.

Our goal is to learn a policy $\pi_\theta(a|s)$ for the MLLM that enables human-like SAM-using for reasoning segmentation, by maximizing the expected reward $\mathbb{E}_{a \sim \pi_\theta(a|s)}[R(s', s, a)]$. By this means, the optimized SAM-Veteran is capable of using SAM to carry out a complete reasoning segmentation workflow covering box generation, iterative mask refinement, and adaptive termination, emulating the intuitive, human-like logic of the process.

## 3.2 MULTI-TASK REINFORCEMENT LEARNING FOR SAM-VETERAN

To accomplish this workflow, SAM-Veteran relies on two key abilities: (1) **textual grounding**, which generates a bounding box around the target, and (2) **mask comprehension**, which evaluates whether the SAM segmentation is satisfactory and, if not, produces refinement points to guide SAM in improving the mask. To equip the MLLM with these capabilities, we propose a Multi-Task Reinforcement Learning Framework for SAM-Veteran, based on GPRO, as illustrated in Figure 2. Specifically, we provide the MLLM with task-specific prompts (detailed in Appendix A.1), prompting it to generate multiple responses (rollouts), each including both the reasoning process and the

final answer for the corresponding task. For each rollout, we compute a task-specific reward, which is then used for GPRO optimization. The details of each task are outlined in the following sections.

### 3.2.1 TASK 1: TEXTUAL GROUNDING

This task reinforces the MLLM's ability to generate a bounding box for the target. Specifically, the MLLM is prompted to locate the object in an image based on a given question. Formally, given a pair of an image $I$ and a corresponding question $Q$, the MLLM receives a grounding-specific prompt $Q^{\mathrm{B}}$, which guides it to generate the reasoning process and bounding box coordinates $b$. We define three types of rewards for this task:

- **Box Reward**: We define our box reward as the combination of box IoU reward $R^{\mathrm{B}}_{\mathrm{IoU}}$ and box $L_1$ reward $R^{\mathrm{B}}_{L_1}$ proposed in Seg-Zero. Given the GT box $b^{\mathrm{GT}} \in \mathbb{R}^4$, each component is defined as,

$$R^{\mathrm{B}}_{\mathrm{IoU}} = \left\{ \begin{array}{ll} 1, & \mathrm{IoU}(b, b^{\mathrm{GT}}) > 0.5 \\ 0, & \mathrm{otherwise,} \end{array} \right. \qquad R^{\mathrm{B}}_{L_1} = \left\{ \begin{array}{ll} 1, & \sum_i |b_i - b_i^{\mathrm{GT}}|/4 < 10 \\ 0, & \mathrm{otherwise.} \end{array} \right.$$

- **SAM Reward** $R^{\mathrm{SAM}}$: We input the image $I$ and the bounding box $b$ into SAM, which produces the segmentation mask $M = \mathrm{SAM}(I, b)$. The SAM reward $R^{\mathrm{SAM}}$ is defined as the IoU between the predicted mask $M$ and the ground truth mask $M^{\mathrm{GT}}$, i.e., $R^{\mathrm{SAM}} = \mathrm{IoU}(M, M^{\mathrm{GT}})$. SAM reward evaluates how well the box supports SAM's segmentation.

- **Formatting Reward**: This reward enforces compliance with the required output format. In particular, the MLLM must encapsulate its reasoning process and final answer within predefined tags (1 score) and provide the answer in a strictly parsable format (1 score). This reward applies uniformly across all tasks.

### 3.2.2 TASK 2: MASK COMPREHENSION

This task trains the MLLM to generate points that serve as additional inputs for SAM to refine the mask, or to terminate the process if the mask is already satisfactory. Specifically, we provide the MLLM with an image overlaid by a green transparent mask $M$ (following SegAgent), predicted by SAM, together with the corresponding question $Q$. The MLLM is then prompted with a refinement-specific prompt $Q^{\mathrm{P}}$, from which it predicts a positive point $p^+$ and/or a negative point $p^-$. The image $I$, the mask $M$, and the refinement points $(p^+, p^-)$, are fed into SAM, which outputs an

Table 1: IoU Imp. Reward.

| $\Delta$ | (-1, 0] | (0, 0.1] | (0.1, 0.5] | (0.5, 1] |
|---|---|---|---|---|
| $R^{\Delta}$ | 0 | 1 | 2 | 3 |

Table 2: Reward combinations in all cases.

| Action $a$ | Good Enough | Need Refinement |
|---|---|---|
| $(p^+, p^-)$ | 0 | |
| (null, null) | $R^{\mathrm{DCS}} + 3$ | $R^{\mathrm{DCS}} + R^{\Delta}$ |
| Others | $R^{\mathrm{ENC}}$ | |

improved mask $M' = \mathrm{SAM}(I, M, p^+, p^-)$. Regarding rewards, in addition to the SAM reward $R^{\mathrm{SAM}} = \mathrm{IoU}(M', M^{\mathrm{GT}})$ and formatting reward, we also incorporate decision reward and an IoU improvement (IoU imp.) reward for this task:

- **Decision Reward** $R^{\mathrm{DCS}}_C$: This reward guides the MLLM to terminate the refinement process at the appropriate time. Given a state $(M, I, Q)$, two cases are considered: (1) the current mask is unsatisfactory and requires further refinement (*Need Refinement*), and (2) the mask is sufficiently accurate and should be accepted (*Good Enough*). The MLLM receives a score of 1 if it produces refinement points ($a \neq$ (null, null)) when the mask is in the *Need Refinement* case, or if it terminates the process ($a =$ (null, null)) when the mask is in the *Good Enough* case. In all other situations, the reward is set to 0.

- **IoU Imp. Reward** $R^{\Delta}$: This reward evaluates how much the IoU is improved by the current action if the mask needs refinement. It is defined based on the IoU change $\Delta = \mathrm{IoU}(M', M^{\mathrm{GT}}) - \mathrm{IoU}(M, M^{\mathrm{GT}})$, as shown in Table 1. Specifically, the MLLM receives the highest score of 3 if the mask shows a significant IoU improvement ($\Delta > 0.5$); the reward is set to 0 if the refinement process degrades the mask ($\Delta \leq 0$).

The overall reward of this task is determined by case–action pairs. Table 2 summarizes the combination of the two rewards for each pair. For consistency, the maximum reward is fixed at 4 in both

the *Need Refinement* and *Good Enough* cases. In addition, an encouraging score of $R^{\text{ENC}} = 2$ is granted when the model generates only one type of point in the *Good Enough* case.

A key challenge in this task lies in explicitly obtaining masks under the two designated scenarios. To collect the data, we first extract ground-truth (GT) bounding boxes for the target objects in the training set and then provide each box–image pair to SAM for segmentation. The resulting masks are categorized as *Need Refinement* if their IoU is low, and as *Good Enough* if their IoU is high. To reduce the impact of imperfect edge annotations in the GT masks and to ensure that the MLLM focuses on substantive segmentation errors, we adopt a modified IoU calculation that excludes near-edge pixels[1]. Under this scheme, masks with $\text{IoU} = 1$ are classified as *Good Enough*, whereas those with $\text{IoU} < 0.9$ are deemed *Need Refinement*. Furthermore, we balance the ratio between these two categories to avoid training bias.

### 3.2.3 TASK 3: AUXILIARY MASK COMPREHENSION

When trained solely on the two tasks introduced above, we observe that the MLLM continues refining the mask indefinitely, even when the mask is already perfect. We hypothesize that this behavior arises from the MLLM's limited ability to interpret masks within images, likely due to insufficient exposure to such data during pretraining. To enhance the model's mask comprehension, we introduce an auxiliary task, where ground-truth (GT) masks are intentionally corrupted with artificial imperfections, including random polygonal inclusions (serve as false positives) and exclusions (serve as false negatives). Specifically, the MLLM is provided with the image overlaid with the imperfect mask $M$ and the corresponding question $Q$. The same refinement-specific question $Q^{\text{P}}$ is used to prompt the model, guiding it to predict a positive point $p^+$ and a negative point $p^-$, corresponding to the centers of the false negative and false positive regions, respectively. To enforce this behavior, we define the following rewards:

- **Decision Reward** $R_A^{\text{DCS}}$: This reward encourages the MLLM to output the appropriate type of point accordingly. Specifically, when a false positive region exists, the model receives a score of 1 for outputting $p^- \in \mathbb{R}^2$. Conversely, if no false positive region is present, it receives a score of 1 for outputting `null`. An analogous reward scheme is applied to the false negative region.

- **Point Reward**: This reward encourages the MLLM to generate points at the correct locations. Specifically, the model receives a score of 1 if the output $p^-$ falls within a false positive region, and an additional score of 1 if its distance from the center of that region is less than $\tau_d = 50$ pixels. An analogous reward scheme is defined for $p^+$.

### 3.3 DYNAMIC SAMPLING

Dynamic Sampling, proposed in DAPO, addresses the issue of gradient vanishing in GRPO. The original dynamic sampling strategy keeps sampling until the batch is fully filled with samples whose accuracy is neither 0 nor 1. We adapt and extend this idea to the three training tasks described above. For the grounding task, we over-sample candidate boxes and then apply Non-Maximum Suppression (NMS) to remove duplicates. This allows multiple objects or locations within the image to be sampled, thereby increasing the diversity of rewards across rollouts and enhancing the effectiveness of the advantage function. For the mask comprehension and auxiliary mask comprehension tasks, actions can be categorized into four cases: $(\texttt{null}, \texttt{null})$, $(p^+, \texttt{null})$, $(\texttt{null}, p^-)$, and $(p^+, p^-)$. We over-sample actions until all four cases are represented. This practice encourages the MLLM to recognize the differences among possible actions. By ensuring a diverse set of actions with varying rewards, optimization becomes more stable and effective.

## 4 EXPERIMENTS

### 4.1 EXPERIMENT SETTINGS

**Implementation Details.** We adopt Qwen2.5-VL-7B-Instruct (Bai et al., 2025) as our base MLLM and SAM2-Large (Ravi et al., 2025) as the segmentation module. For training, we configure the

---

[1]We employ the modified IoU for all IoU-based mask rewards ($R^{\text{SAM}}$ and $R^{\Delta}$) during training, while original IoU is retained in evaluation for fair comparison. The detailed computation is in Appendix A.3.

Table 3: We compare IoU (%) of different MLLM-based methods (7B version) across both in-domain and out-of-domain datasets.

| Method | Out-of-Domain | | | | In-Domain | | |
| --- | --- | --- | --- | --- | --- | --- | --- |
| | ReasonSeg val | | ReasonSeg test | | RefCOCO testA | RefCOCO+ testA | RefCOCOg test |
| | gIoU | cIoU | gIoU | cIoU | cIoU | cIoU | cIoU |
| Qwen2.5-VL+SAM2 | 57.2 | 41.4 | 53.0 | 48.1 | 76.1 | 71.4 | 64.8 |
| *Supervised Finetuning* | | | | | | | |
| LISA (Lai et al., 2024) | 53.6 | 52.3 | 48.7 | 48.8 | 76.5 | 67.4 | 68.5 |
| VISA (Yan et al., 2024) | 52.7 | 57.8 | — | — | 75.7 | 64.8 | 66.4 |
| PixelLM (Ren et al., 2024b) | — | — | — | — | 76.5 | 71.7 | 70.5 |
| PerceptionGPT (Pi et al., 2024) | — | — | — | — | 78.6 | 73.9 | 71.7 |
| GSVA (Xia et al., 2024) | 50.5 | 56.4 | — | — | 78.9 | 69.6 | 73.3 |
| *Reinforcement Learning* | | | | | | | |
| POPEN (Zhu et al., 2025a) | 60.2 | 64.5 | — | — | 79.9 | 74.4 | **74.6** |
| SegAgent (Zhu et al., 2025b) | 33.0 | 25.4 | 33.5 | 31.3 | 80.3 | 75.5 | **74.6** |
| Seg-Zero (Liu et al., 2025a) | 62.6 | 62.0 | 57.5 | 52.0 | 80.3 | 76.2 | 72.6 |
| SAM-R1 (Huang et al., 2025) | 64.0 | 55.8 | 60.2 | 54.3 | 79.2 | 74.7 | 73.1 |
| SAM-Veteran | **68.2** | **67.3** | **62.6** | **56.1** | **80.8** | **76.6** | 73.4 |

batch size to 16 alongside a rollout number of 8. AdamW (Loshchilov & Hutter, 2017) is exploited as the optimizer with the learning rate $10^{-6}$, weight decay 0.01, and KL coefficient 0.005. To stabilize the training, we adopt global batch normalization from REINFORCE++ (Hu et al., 2025) instead of local standard deviation in GRPO. The model is trained for one episode with the verl (Sheng et al., 2025) framework on eight 96GB GPUs, taking about 30 hours. To balance effectiveness and efficiency in evaluation, we limit the refinement process to a maximum of 3 steps unless specified. Following Seg-Zero and SAM-R1, we resize all images to $840 \times 840$ for both training and evaluation. In addition, during each refinement step, we supply SAM with the initial bounding box obtained from grounding. More implementation details are provided in Appendix A.2.

**Datasets and Evaluation Metrics.** Regarding datasets, we use RefCOCOg (Yu et al., 2016) to train our SAM-Veteran (more details in Appendix A.4) and evaluate it on RefCOCO(+/g) (in-domain) and ReasonSeg (Lai et al., 2024) (out-of-domain). Following Seg-Zero, we use gIoU and cIoU as the evaluation metrics, where gIoU is the average of per-image Intersection-over-Unions (IoUs), and the cIoU is the cumulative intersection over the cumulative union.

## 4.2 COMPARISONS

We compare SAM-Veteran with both SFT-based and RL-based models, with results summarized in Table 3. For Qwen2.5-VL, segmentation masks are obtained by directly passing its predicted bounding boxes into SAM. As shown in the table, SAM-Veteran outperforms all SFT-based methods on both out-of-domain and in-domain benchmarks. Compared with Seg-Zero, it achieves stronger results across all benchmarks, primarily because our framework integrates SAM's segmentation outputs into the RL reward and enables iterative mask refinement. Although SegAgent reports higher scores on the RefCOCOg test set, its reliance on SFT with point trajectories limits its generalization ability, leading to poor performance on out-of-domain data; furthermore, it requires more refinement steps and lacks adaptive termination. POPEN also achieves the best performance on RefCOCOg test; however, it relies on a well-trained PixelLM as its initialization and adopts a relatively complex multi-stage training pipeline. The previous out-of-domain SOTA, SAM-R1, attains strong performance by adopting GRPO with SAM reward as guidance. However, unlike SAM-Veteran, it does not support iterative refinement, resulting in a performance gap. Overall, on in-domain datasets, SAM-Veteran achieves performance comparable to or exceeding existing methods in terms of IoU, while on out-of-domain datasets, it consistently surpasses all baselines by a clear margin, underscoring the generalization benefits of our reinforcement learning framework.

## 4.3 ABLATION STUDY

**Iterative Mask Refinement.** Figure 3 illustrates the trends in IoU across refinement steps. As shown, SAM-Veteran demonstrates an obvious trend of improving the IoU compared to the initial

Figure 3: Trends of IoU ($\Delta$) and termination ratio over refinement iterations.

Table 4: Ablation study on three training tasks: Textual Grounding (*TG*), Mask Comprehension (*MC*), and Auxiliary (*A*). We report the IoU along with the termination behavior of the models.

| TG | MC | A | ReasonSeg val | ReasonSeg test | RefCOCO testA | RefCOCO+ testA | RefCOCOg test | Avg. | Termination |
|---|---|---|---|---|---|---|---|---|---|
| Qwen+SAM2 | | | 57.2 | 53.0 | 76.1 | 71.4 | 64.8 | 64.4 | Arbitrary |
| ✓ | | | 62.4 | 62.1 | 79.3 | 75.3 | 72.2 | 70.3 | Arbitrary |
| ✓ | ✓ | | 67.4 | 62.5 | 80.6 | 76.2 | **73.6** | 72.1 | Never |
| ✓ | ✓ | ✓ | **68.2** | **62.6** | **80.8** | **76.6** | 73.4 | **72.2** | **Adaptive** |

mask generated from a bounding box (step 0), with particularly notable gains on out-of-domain datasets. In addition, we report the ratio of terminated samples at each step. The figure shows that this termination ratio increases with iterative refinement, indicating that SAM-Veteran progressively considers more masks satisfactory and is able to adaptively terminate the process when appropriate. For the original Qwen model, the IoU declines throughout the process, indicating insufficient mask comprehension to support effective refinement.

**Multi-Task Training.** Table 4 presents the impact of the three training tasks. For Qwen and our model trained solely on the textual grounding task, we report results based only on the predicted bounding boxes. Our pure grounding model outperforms Qwen by a large margin, but both exhibit arbitrary termination behavior. Incorporating mask comprehension substantially improves accuracy; however, it leads to indefinite refinement in the absence of an effective termination policy. Introducing the auxiliary task further strengthens semantic understanding and enables adaptive termination. When all three tasks are combined, the model achieves the highest overall IoU while maintaining an adaptive refinement termination strategy.

**Reward Design.** Table 5 shows the results of the ablation study on our reward design. Specifically, We study the model trained without SAM reward $R^{\text{SAM}}$, decision reward $R_*^{\text{DCS}}$ ($R_C^{\text{DCS}}$ and $R_A^{\text{DCS}}$), or IoU improvement reward $R^{\Delta}$. Besides, we also replace $R^{\Delta}$ with a hard version $R_h^{\Delta} = 3 \cdot \mathbb{1}_{\Delta>0}$. As the results show, removing either component leads to consistent performance drops across datasets, highlighting their complementary roles in guiding effective refinement. Moreover, $R_h^{\Delta}$ yields weaker results than the original design, confirming that $R^{\Delta}$ is more effective.

**Dynamic Sampling and Chain of Thought.** The ablation results in Table 6 show that both dynamic sampling (DS) and chain of thought (CoT) contribute to SAM-Veteran's performance. Removing DS leads to noticeable drops across most benchmarks, indicating its role in enhancing robustness. Meanwhile, removing CoT in all tasks also degrades performance, particularly on ReasonSeg, while the variant without CoT in mask-related tasks performs better but still lags behind the full model. These results confirm that DS and CoT are complementary and jointly important for effectiveness.

Table 5: Ablation study on reward design, including removing SAM reward $R^{\text{SAM}}$, decision reward $R_*^{\text{DCS}}$, and IoU improvement reward $R^{\Delta}$, and replacing $R^{\Delta}$ with a hard version $R_h^{\Delta}$.

| Model | ReasonSeg val | ReasonSeg test | RefCOCO testA | RefCOCO+ testA | RefCOCOg test | Avg. |
|---|---|---|---|---|---|---|
| SAM-Veteran | **68.2** | **62.6** | 80.8 | **76.6** | **73.4** | **72.2** |
| w/o $R^{\text{SAM}}$ | 67.0 | 60.7 | 80.4 | 76.1 | 72.2 | 71.3 |
| w/o $R_*^{\text{DCS}}$ | 64.1 | 60.6 | 80.4 | 75.7 | 72.1 | 70.6 |
| w/o $R^{\Delta}$ | 64.1 | 60.2 | 79.4 | 75.9 | 72.3 | 70.4 |
| $R^{\Delta} \to R_h^{\Delta}$ | 67.5 | 61.9 | **81.0** | 76.2 | 73.0 | 71.9 |

Table 6: Ablation study on dynamic sampling (DS) and chain of thought (CoT).

| Model | ReasonSeg val | ReasonSeg test | RefCOCO testA | RefCOCO+ testA | RefCOCOg test | Avg. |
|---|---|---|---|---|---|---|
| SAM-Veteran | **68.2** | **62.6** | **80.8** | 76.6 | **73.4** | **72.2** |
| w/o DS | 66.8 | 60.8 | 79.3 | 75.4 | 71.8 | 70.8 |
| w/o CoT | 64.4 | 60.1 | 80.0 | 75.5 | 73.0 | 70.6 |
| w/o Mask CoT | 66.1 | 61.5 | 80.3 | **76.8** | 73.3 | 71.6 |

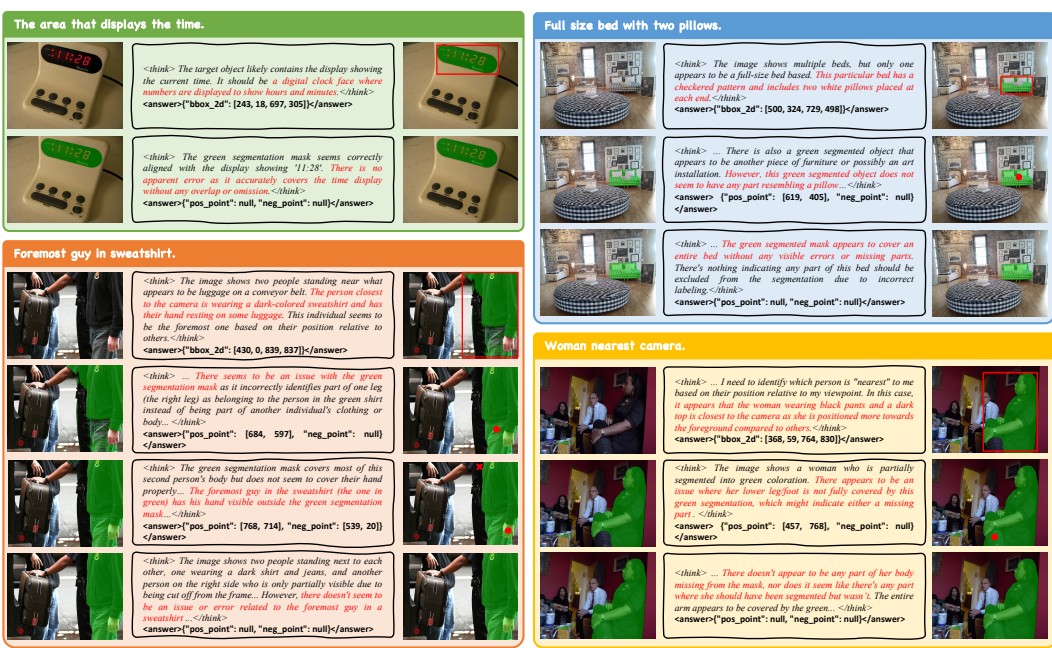

Figure 4: Examples of the workflow with SAM-Veteran. In each example, the first column presents the visual input at each step, the second column shows the text response, and the third column displays the SAM prediction given the boxes/points. Positive/Negative points are shown as ●/✕.

**Visualization.** Figure 4 presents examples of the segmentation workflow performed by SAM-Veteran. In each case, the first column shows the visual input at each step, the second column displays the SAM-Veteran response, and the third column provides the SAM prediction given the boxes/points. We illustrate three representative scenarios, corresponding to 0, 1, and 2 refinement steps. These visualizations demonstrate that SAM-Veteran is capable of human-like usage of SAM, including providing the target box, identifying segmentation errors, and terminating once the mask is satisfactory. Moreover, the process is reasoning-driven, as the model generates a plausible chain of thought to guide the final prediction. More visualizations are in Appendix B.6.

## 5  CONCLUSION

We propose SAM-Veteran, an MLLM-based agent designed to perform human-like SAM usage for reasoning segmentation. Given an image and a text query, SAM-Veteran executes a complete workflow: (1) generating a bounding box as input to SAM for initial mask prediction, (2) iteratively producing refinement points that serve as additional inputs for mask refinement, and (3) adaptively terminating the process once the mask is deemed satisfactory. To equip the MLLM with these capabilities for the workflow, we develop a multi-task reinforcement learning framework that explicitly rewards accurate bounding textual grounding and mask comprehension. Quantitative evaluations and qualitative visualizations demonstrate that SAM-Veteran achieves state-of-the-art performance on both in-domain and out-of-domain benchmarks, while also exhibiting human-like behavior in SAM-based reasoning segmentation.

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

APPENDIX CONTENTS

## A    REPRODUCIBILITY: MORE IMPLEMENTATION DETAILS

### A.1    PROMPTS USED

We illustrate our prompts for textual grounding, mask comprehension, and auxiliary mask comprehension in Figure 5.

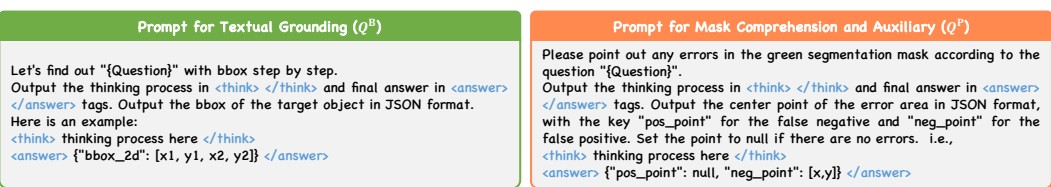

Figure 5: Prompts used for textual grounding, mask comprehension, and the auxiliary task.

### A.2    MORE CONFIGURATION DETAILS

In our dynamic sampling strategy for bounding boxes, we apply Non-Maximum Suppression (NMS) with an IoU threshold of 0.8 to eliminate duplicates. For refinement points, we generate samples in counts of $(1, 2, 2, 1)$ corresponding to the cases $(\texttt{null}, \texttt{null})$, $(p^+, \texttt{null})$, $(\texttt{null}, p^-)$, and $(p^+, p^-)$, respectively. The pseudocode for dynamic sampling of bounding boxes and refinement points is provided in Algorithm 1 and Algorithm 2. To accelerate training, dynamic sampling is disabled after 300 iterations. Following DAPO, we increase the upper clipping bound to 0.28 to encourage exploration. During evaluation, a repetition penalty of 1.1 is applied to reduce redundant token generation. Full configurations can be found in Figure 6.

---

**Algorithm 1:** Dynamic Sampling for Box

**Input:** Actor model $M$, image $I$, text prompt $P$, number of rollouts $R$, maximum attempts $A$, maximum error count $E$, IoU threshold $\tau$
**Output:** Valid rollouts $\mathcal{R}$
$\mathcal{R} \leftarrow \varnothing, \mathcal{B} \leftarrow \varnothing, n_e \leftarrow 0$;
**for** $a \leftarrow 1$ **to** $A - 1$ **do**
  $\mathcal{C} \leftarrow M(I, P), \mathcal{B}' \leftarrow \mathcal{B}$;
  $n_p \leftarrow |\mathcal{B}|, r_{\text{idx}} \leftarrow \{-1\}_{i=1}^{|\mathcal{B}|}$;
  **foreach** *rollout $r$ with index $j$ in $\mathcal{C}$* **do**
    **if** $|\mathcal{R}| = R$: **return** $\mathcal{R}$;
    **if** $\neg$ *correctFormat(r)* **and** $n_e < E$ **:**
      $\mathcal{R} \leftarrow \mathcal{R} \cup \{r\}, n_e \leftarrow n_e + 1$;
      **continue**;
    $\mathcal{B}' \leftarrow \mathcal{B}' \cup \{\text{getBox}(r)\}$;
    $r_{\text{idx}} \leftarrow r_{\text{idx}} \cup \{j\}$;
  **foreach** $k \in NMS(\mathcal{B}', \tau)[: R]$ **do**
    **if** $k \geq n_p$ **and** $|\mathcal{B}| < R$ **:**
      $\mathcal{B} \leftarrow \mathcal{B} \cup \{\mathcal{B}'[k]\}$;
      $\mathcal{R} \leftarrow \mathcal{R} \cup \{\mathcal{C}[r_{\text{idx}}[k]]\}$;
**if** $R - |\mathcal{R}| > 0$ **:**
  $\mathcal{C} \leftarrow M(I, P), \mathcal{R} \leftarrow \mathcal{R} \cup \mathcal{C}[: (R - |\mathcal{R}|)]$;
**return** $\mathcal{R}$;

---

**Algorithm 2:** Dynamic Sampling for Point

**Input:** Actor model $M$, image $I$, text prompt $P$, number of rollouts $R$, maximum attempts $A$
**Output:** Valid rollouts $\mathcal{R}$
$\mathcal{R} \leftarrow \varnothing$;
$n_e \leftarrow 0$;
cases $\leftarrow \{\text{error} : 2, \text{terminate} : 1, \text{positive} : 2, \text{negative} : 2, \text{both} : 1\}$;
**for** $a \leftarrow 1$ **to** $A - 1$ **do**
  $\mathcal{C} \leftarrow M(I, P)$;
  **foreach** *rollout $r \in \mathcal{C}$* **do**
    **if** $|\mathcal{R}| = R$ **:**
      **return** $\mathcal{R}$;
    $c \leftarrow \text{getCase}(r)$;
    **if** *cases[c] > 0* **:**
      $\mathcal{R} \leftarrow \mathcal{R} \cup \{r\}$;
      cases$[c] \leftarrow$ cases$[c] - 1$;
**if** $R - |\mathcal{R}| > 0$ **:**
  $\mathcal{C} \leftarrow M(I, P)$;
  $\mathcal{R} \leftarrow \mathcal{R} \cup \mathcal{C}[: (R - |\mathcal{R}|)]$;
**return** $\mathcal{R}$;

---

```yaml
data:
    max_prompt_length: 1300
    max_response_length: 1300
    tasks:
      - task: "Textual Grounding"
        train_files: data/refCOCOg_9k_840_mask
        repeat: 1
      - task: "Mask Comprehension"
        train_files: data/refCOCOg_600_700_840_mask
        repeat: 5
      - task: "Auxiliary"
        train_files: data/refCOCOg_9k_840_mask
        repeat: 1
algorithm:
  adv_estimator: grpo
  use_batch_std: true
worker:
  actor:
    global_batch_size: 16
    use_kl_loss: true
    kl_loss_coef: 5.0e-3
    clip_low: 0.2
    clip_high: 0.28
    optim:
      optimizer: AdamW
      lr: 1.0e-6
      weight_decay: 1.0e-2
  rollout:
    temperature: 1.0
    n: 8
    dynamic_sample:
      max_try: 3
      stop_iter: 300
      tasks:
        - task: "Textual Grounding"
          meta_infos:
            iou_thr: 0.8
            max_error_cnt: 2
        - task: "Mask Comprehension"
          meta_infos:
            required_cases: [1, 2, 2, 1]
            max_error_cnt: 2
        - task: "Auxiliary"
          meta_infos:
            required_cases: [1, 2, 2, 1]
            max_error_cnt: 2
  reward:
    tasks:
      - task: "Textual Grounding"
        reward_list: ["formatting reward","sam reward","box reward"]
      - task: "Mask Comprehension"
        reward_list: ["formatting reward","sam reward","decision reward","iou imp. reward"]
      - task: "Auxiliary"
        reward_list: ["formatting reward","decision reward","point reward"]
trainer:
  total_episodes: 1
```

Figure 6: Full configurations for the training of SAM-Veteran.

### A.3 IoU Excluding Near-Edge Pixels

As mentioned in SegAgent, the masks in refCOCO(+/g) (Yu et al., 2016) were annotated with polygons, resulting in imperfect alignment with boundaries. So we refine the IoU computation in calculating rewards by excluding pixels near the object boundaries, as the Python code shown in Figure 7. In addition, we also visualize the near-edge pixels computed from our algorithm in Figure 8.

### A.4 Training Data

For the grounding task, we follow Seg-Zero and use the same 9,000 samples from the full training set of RefCOCOg. For the mask comprehension task, we also use these 9,000 samples, obtaining SAM predictions from the ground-truth bounding boxes. We then compute the edge-pixel-excluded IoU between the SAM masks and ground-truth masks. 600 samples with $\text{IoU} = 1$ are treated as *Good Enough*, while around 700 samples with $\text{IoU} < 0.9$ are categorized as *Need Refinement*. For the auxiliary mask comprehension task, we randomly augment the 9,000 ground-truth masks by adding polygon-region inclusion or exclusion, each applied independently with probability 0.5. The examples of data in the mask-related tasks are shown in Figure 9. To balance the data across tasks, we upsample the mask comprehension data by repeating it five times during training.

```python
import numpy as np
import cv2

def compute_iou(pred_mask: np.ndarray, gt_mask: np.ndarray,
↪   ignore_edge: int = 20) -> float:
    pred_mask = pred_mask.astype(bool)
    gt_mask   = gt_mask.astype(bool)
    # Create ignore mask
    if ignore_edge > 0:
        # Use Canny to detect edges
        edges = cv2.Canny(gt_mask.astype(np.uint8) * 255, 100, 200)
        kernel = np.ones((ignore_edge, ignore_edge), np.uint8)
        ignore_mask = cv2.dilate(edges, kernel,
        ↪   iterations=1).astype(bool)
    else:
        ignore_mask = np.zeros_like(gt_mask, dtype=bool)
    # Apply ignore mask
    valid_pred = np.logical_and(pred_mask, ~ignore_mask)
    valid_gt   = np.logical_and(gt_mask, ~ignore_mask)
    intersection = np.logical_and(valid_pred, valid_gt).sum()
    union        = np.logical_or(valid_pred, valid_gt).sum()
    if union == 0:
        return 1  # avoid NaN, if both masks are empty
    return intersection / union
```

Figure 7: Python code to compute IoU excluding pixels near the object boundaries.

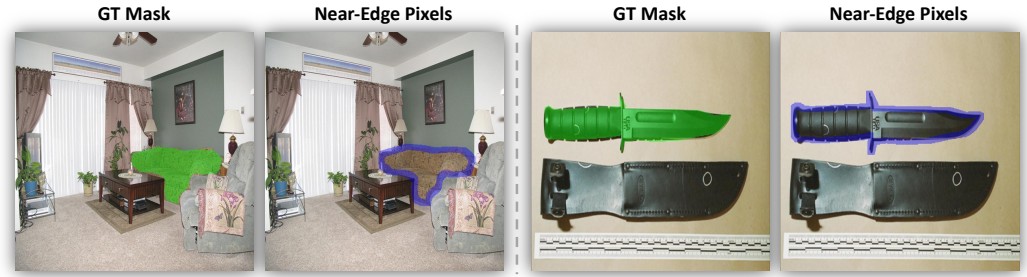

Figure 8: Visualization of GT mask (green) and the corresponding near-edge pixels (blue). We exclude the near-edge pixels in IoU computation for training process only.

## B    MORE EXPERIMENTS AND ANALYSIS

### B.1    MORE REASONING SEGMENTATION BENCHMARKS

To further demonstrate the effectiveness of our SAM-Veteran, we evaluated it on two more reasoning segmentation benchmarks, i.e., MMR (Jang et al., 2025) and MUSE (Ren et al., 2024b). For MMR, we adopt the object-only version for fair comparison. Because a query in MMR may refer to multiple objects, we merge the corresponding instance masks and treat the merged mask as the ground-truth segmentation. For MUSE (test set), its original setting involves multi-target and multi-referring segmentation, which differs from the widely used benchmarks. To align with our evaluation protocols, we convert each sample into a classic triplet of image, single-object mask, and mask caption. We then feed the image and mask caption into the model and compare the predicted mask with the single-object ground-truth mask. Table 7 presents the results of different models on the two benchmarks. Qwen+SAM2 provides the baseline performance on both datasets. Seg-Zero delivers a substantial improvement over this baseline. In contrast, SegAgent performs poorly on MUSE and nearly fails on MMR, reflecting the limited generalization ability introduced by its SFT-

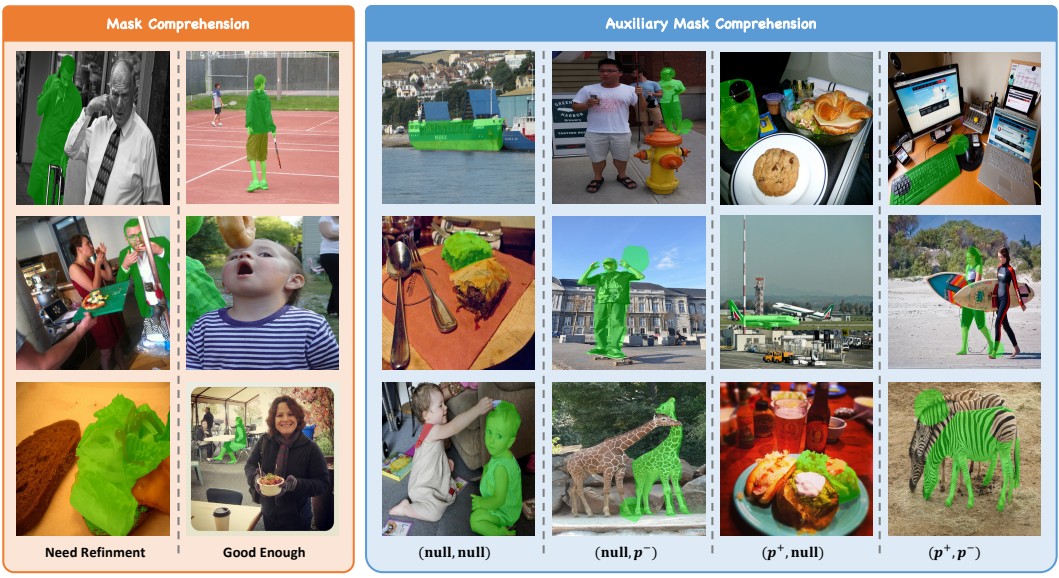

Figure 9: Examples of visual inputs for the task of mask comprehension and the task of auxiliary mask comprehension.

Table 7: Results on MMR and MUSE.

| Method | MMR (Jang et al., 2025) | | MUSE (Wang et al., 2025c) | |
|---|---|---|---|---|
| | gIoU | cIoU | gIoU | cIoU |
| Qwen+SAM2 | 33.18 | 26.63 | 43.28 | 45.79 |
| Seg-Zero (Liu et al., 2025a) | 37.91 | 29.44 | 52.16 | 54.38 |
| SegAgent (Zhu et al., 2025b) | 13.34 | 16.56 | 37.29 | 43.48 |
| SAM-Veteran | 40.38 | 30.74 | 53.63 | 57.42 |

based training. Our SAM-Veteran further surpasses Seg-Zero by a clear margin on both benchmarks, demonstrating the effectiveness of our RL-based multi-task training framework.

## B.2 GENERAL MLLM BENCHMARKS

To substantiate our claim that the proposed RL framework mitigates catastrophic forgetting of general reasoning ability, we evaluate SAM-Veteran—along with several baseline models—on standard general-purpose MLLM benchmarks (results shown in Table 8). As observed, the RL-based methods, Seg-Zero and our SAM-Veteran, maintain performance comparable to their respective base MLLM (Qwen2.5-VL), demonstrating that RL training preserves general reasoning capability. In contrast, the SFT-based model SegAgent exhibits a clear decline in performance on general vision-language benchmarks relative to its base model, Qwen-VL-Chat (Bai et al., 2023), indicating significant loss of generalization ability. These results confirm that SFT-based training is prone to catastrophic forgetting, whereas RL-based methods effectively avoid this issue.

## B.3 MODEL SCALABILITY

To evaluate the scalability of our method, we scale the MLLM from 7B to 32B and conduct experiments on Qwen2.5-VL-32B using the same settings as the 7B model. To further test scalability in the refinement step, we increase the maximum refinement steps from 3 to 5 for the 32B model. The results of the 32B variant across different datasets are presented in Table 9. As observed, the 32B version of SAM-Veteran achieves further improvements on most in-domain and out-of-domain datasets, confirming the scalability of our approach.

Table 8: Results of different models on general MLLM benchmarks. * means re-evaluation by in our environment.

| Dataset | Qwen2.5-VL | Qwen2.5-VL* | Seg-Zero* | SAM-Veteran | Qwen-VL-Chat* | SegAgent* |
|---|---|---|---|---|---|---|
| ***OCR-Related Understanding*** | | | | | | |
| SEED-Bench-2-Plus (Li et al., 2024) | 70.4 | 69.6 | 69.5 | 69.5 | 44.6 | 9.7 |
| TextVQA$_{val}$ (Singh et al., 2019) | 84.9 | 85.3 | 85.4 | 84.2 | 60.2 | 1.29 |
| ***General Visual Question Answering*** | | | | | | |
| MMStar (Chen et al., 2024a) | 63.9 | 59.9 | 61.3 | 60.5 | 29.0 | 5.3 |
| MME$_{sum}$ (Fu et al., 2025) | 2347 | 2303 | 2286 | 2328 | 1834 | 753 |
| MUIRBench (Wang et al., 2025a) | 59.6 | 58.3 | 57.4 | 59.2 | 27.9 | 12.23 |
| BLINK (Fu et al., 2024) | 56.4 | 54.3 | 55.3 | 54.0 | 14.4 | 4.42 |
| RealWorldQA (xAI, 2024) | 68.5 | 67.8 | 68.9 | 66.0 | 45.8 | 1.57 |

Table 9: Results of SAM-Veteran of 7B and 32B.

| Model | ReasonSeg val | | ReasonSeg test | | RefCOCO testA | RefCOCO+ testA | RefCOCOg test |
|---|---|---|---|---|---|---|---|
| | gIoU | cIoU | gIoU | cIoU | cIoU | cIoU | cIoU |
| SAM-Veteran-7B | 68.2 | 67.3 | 62.6 | 56.1 | 80.8 | 76.6 | 73.4 |
| SAM-Veteran-32B | 72.3 | 70.0 | 62.9 | 58.2 | 80.4 | 77.4 | 73.4 |

## B.4 HYPERPARAMETER ANALYSIS

We conduct more analysis on hyperparamters as follows, with the results shown in Table 10.

**Task 1** $R_{\text{IoU}}^{\text{B}}$. We explore different designs of $R_{\text{IoU}}^{\text{B}}$ in Task 1. Specifically, we evaluate several hard-threshold settings—0.3, 0.7, and the baseline 0.5—as well as a soft variant defined as $R_{\text{IoU}}^{\text{B}} = \text{IoU}(b, b^{\text{GT}})$. The results indicate that the soft formulation performs worse than all hard-threshold versions, and among the hard thresholds, 0.5 yields the best performance.

**Task 2 Reward.** First, we replace $R^{\Delta}$ in Task 2 with a linear variant whose maximum reward is 3, increasing linearly from 0 with respect to the IoU change $\Delta$, i.e., $R^{\Delta} = 3 \cdot \text{ReLU}(\Delta)$. Second, we set the encouraging score to $R^{\text{ENC}} = 0$. Both modifications lead to a slight decrease in performance compared to the baseline.

**Task 3** $\tau_d$. We experiment with different values of $\tau_d$ in Task 3—10, 30, and the baseline 50. The results show that $\tau_d = 50$ achieves the best overall performance.

**Training Rollout.** For the training hyperparameters, we experiment with different numbers of rollouts in GRPO—4, 16, and the baseline 8. The results show that using 8 rollouts yields the best overall performance.

## B.5 INFERENCE COMPLEXITY

We compare the inference-time efficiency of different models using two metrics: the average number of MLLM inference steps and the average time cost per sample on the RefCOCO testA dataset. The results are shown in Table 11. For Qwen, we report the results of generating boxes for SAM. For

Table 10: Results of different hyperparameter configurations.

| Parameter | Value | ReasonSeg val | ReasonSeg test | RefCOCO testA | RefCOCO+ testA | RefCOCOg test | Avg. |
|---|---|---|---|---|---|---|---|
| | Baseline | 68.2 | 62.6 | 80.8 | 76.6 | 73.4 | 72.3 |
| Task 1 $R_{\text{IoU}}^{\text{B}}$ | Hard 0.7 | 65.7 | 61.2 | 80.5 | 75.9 | 73.0 | 71.3 |
| | Hard 0.3 | 67.1 | 61.2 | 80.6 | 77.3 | 72.5 | 71.7 |
| | Soft | 65.9 | 60.0 | 80.2 | 75.7 | 72.0 | 70.7 |
| Task 2 Reward | $R^{\Delta} = 3 \cdot \text{ReLU}(\Delta)$ | 68.1 | 61.5 | 80.2 | 77.0 | 73.2 | 72.0 |
| | $R^{\text{ENC}} = 0$ | 66.8 | 62.0 | 80.2 | 76.6 | 72.9 | 71.7 |
| Task 3 $\tau_d$ | 10 | 67.3 | 62.1 | 80.3 | 76.6 | 72.0 | 71.6 |
| | 30 | 68.3 | 61.8 | 79.9 | 76.4 | 72.8 | 71.8 |
| Rollout | 4 | 66.5 | 62.0 | 79.9 | 76.1 | 72.0 | 71.3 |
| | 16 | 66.4 | 61.9 | 80.3 | 76.7 | 73.4 | 71.7 |

Table 11: Inference cost comparison of different MLLMs.

| Method | Inference Backend | Average Step | Average Time (s) |
|---|---|---|---|
| Qwen+SAM2 | Transformers | 1 | 3.11 |
| SegZero (Liu et al., 2025a) | Transformers | 1 | 3.43 |
| SegAgent (Zhu et al., 2025b) | Transformers | 7 | 8.95 |
| SAM-Veteran | Transformers | 2.08 | 5.09 |
| | vLLM | 2.21 | 2.47 |

Seg-Zero, the MLLM outputs both the bounding boxes and the points for SAM in a single step, whereas SegAgent adopts a fixed number of 7 refinement iterations for mask prediction.

As shown in the results, Qwen and Seg-Zero finish the task in a single step (about 3s), but their segmentation performance is inferior to that of multi-step refinement methods, as evidenced in Table 1. SegAgent, on the other hand, requires a fixed 7-step MLLM inference pipeline (about 9s), leading to low efficiency. Our SAM-Veteran achieves stronger performance with substantially fewer steps ($< 2.5$ steps on average, about 5s each sample). Our method is slower per sample per step than SegAgent because CoT introduces more response tokens, resulting in improved performance at the cost of extra time. Nevertheless, our method strikes a more favorable balance between segmentation quality and inference efficiency. Furthermore, we improve efficiency by replacing the Transformers backend with the vLLM (Kwon et al., 2023) backend, which significantly reduces the inference time—approximately cutting the time consumption in half. This optimization is applied to all evaluations in our experiments.

### B.6 MORE VISUALIZATION

More visualization of SAM-Veteran is provided in Figure 10. For each example, we show the original image, the initial SAM output from a bounding box, and the masks refined iteratively with points. As the figure demonstrates, SAM-Veteran can accurately locate targets based on the posed question. When the initial segmentation is imperfect, it identifies the error regions, and with additional points as guidance, SAM produces progressively improved masks. Remarkably, the model can capture not only obvious omissions, such as boxes on a shelf, but also subtle, hard-to-detect errors, like imperfect segmentation on a fork and the lawn mower.

As SegAgent also adopts a multi-step mask-refinement framework for reasoning segmentation, we compare its prediction workflow with that of our SAM-Veteran in Figure 11, using both in-domain and out-of-domain examples. As illustrated, SegAgent performs a fixed seven refinement steps and sometimes generates ineffective or irrelevant points. In contrast, SAM-Veteran consistently generates accurate bounding boxes and refinement points, while also being able to dynamically determine when to terminate the procedure. These qualitative results further highlight the advantages of our method over SegAgent.

### B.7 FAILURE CASES

We present several failure cases of SAM-Veteran in Figure 12, categorized into four representative types in different colors. The blue examples illustrate grounding deficiencies, where the model either localizes the wrong target or fails to capture certain instances. The orange examples highlight confusion with green objects or regions in the image, which resemble the green mask. The green examples demonstrate cases where the model generates redundant or wrong points. Finally, the yellow example reveals a flaw in the reasoning process, where the model incorrectly interprets the green mask as part of a human outfit.

## C LIMITATIONS

While SAM-Veteran is capable of performing general human-like SAM usage, it does not yet support fully free-form behavior. For example, it cannot generate new bounding boxes after the initial step or revoke previous actions when the mask quality degrades. Enabling such capabilities would

require a more sophisticated action space and additional data, which we leave for future work. Moreover, following SegAgent, we overlay the mask on the image as input to the MLLM. This changes the object's color and may lead to performance degradation on color-sensitive queries. In future work, we plan to investigate presenting segmentation results to the MLLMs in a more effective way.

## D  USE OF LARGE LANGUAGE MODELS

During the preparation of this manuscript, we employed a Large Language Model (ChatGPT) solely as a writing assistant. Its use was restricted to **refining grammar, improving sentence structure, and enhancing the clarity and readability of the text**. All methods, claims, experimental results, and conclusions were conceived and developed exclusively by the authors.

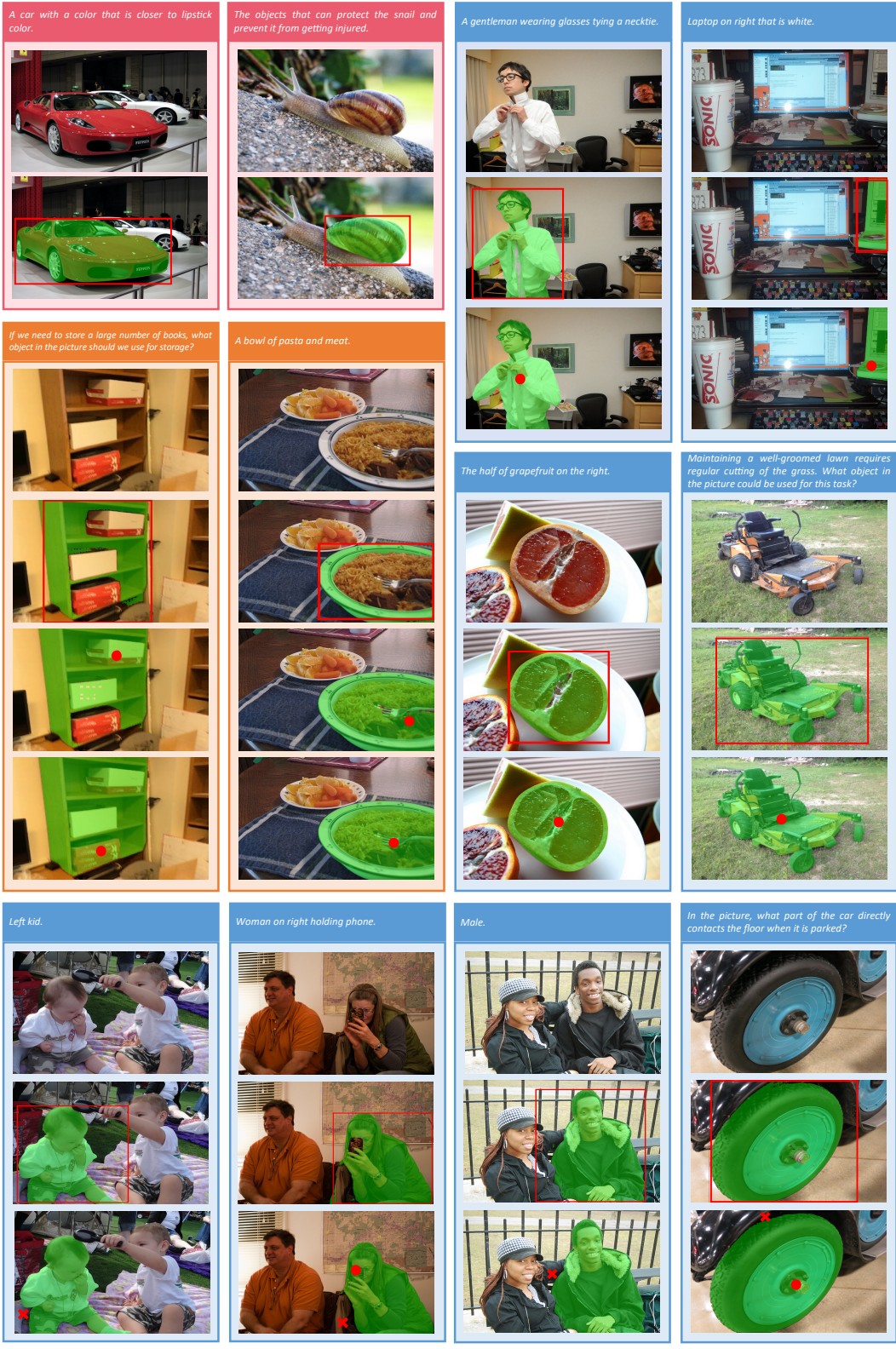

Figure 10: Examples of the segmentation workflow performed by SAM-Veteran. Each example shows the original image, the initial SAM output from a bounding box, and the masks iteratively refined with points. Different number of refinement steps (0, 1, and 2) are in different colors.

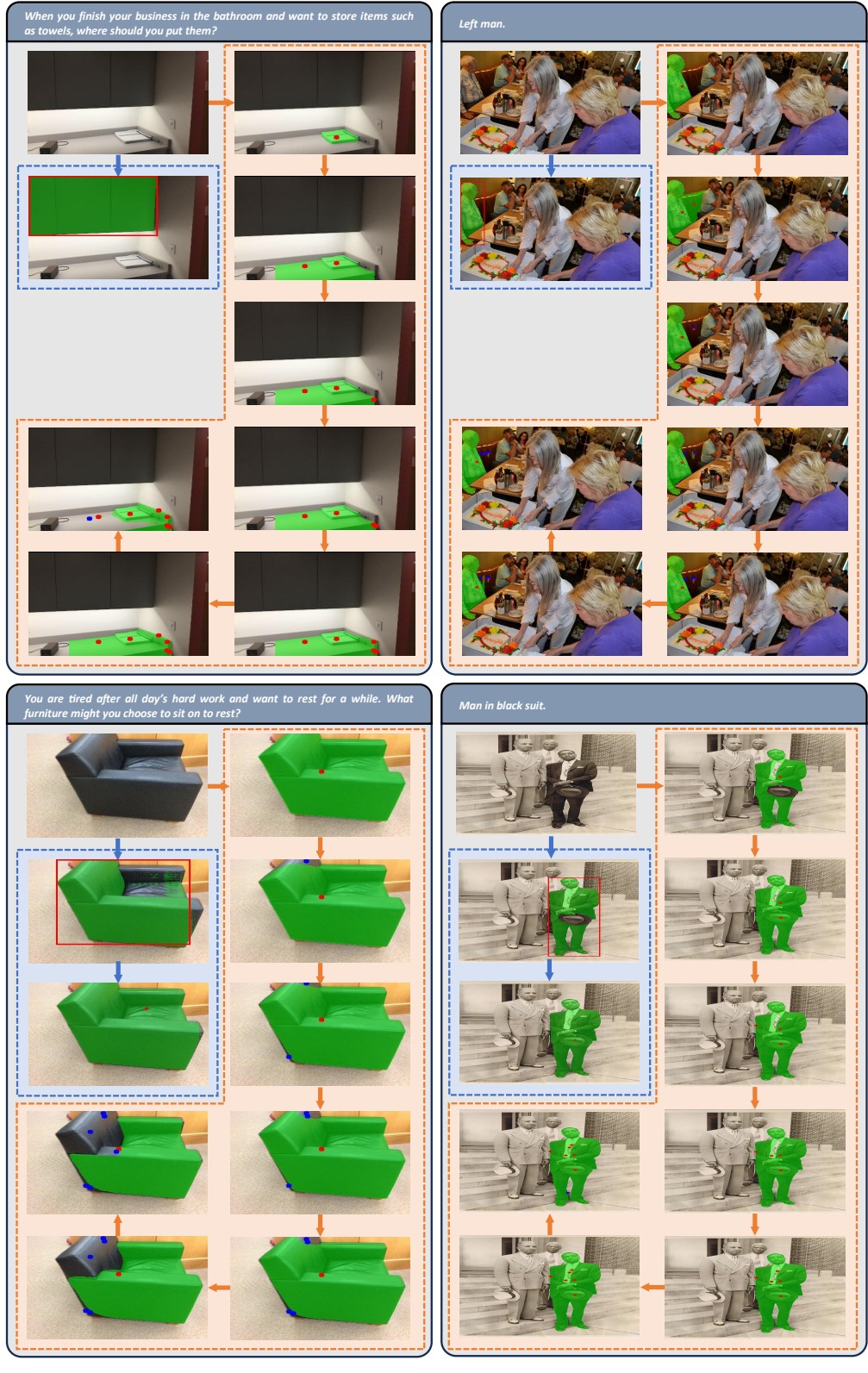

Figure 11: Comparison of the segmentation workflows of SAM-Veteran and SegAgent. The blue and orange flows correspond to SAM-Veteran and SegAgent, respectively. The two examples on the left are out-of-domain cases, while the two on the right are in-domain. Negative points of SegAgent are shown as blue dots •.

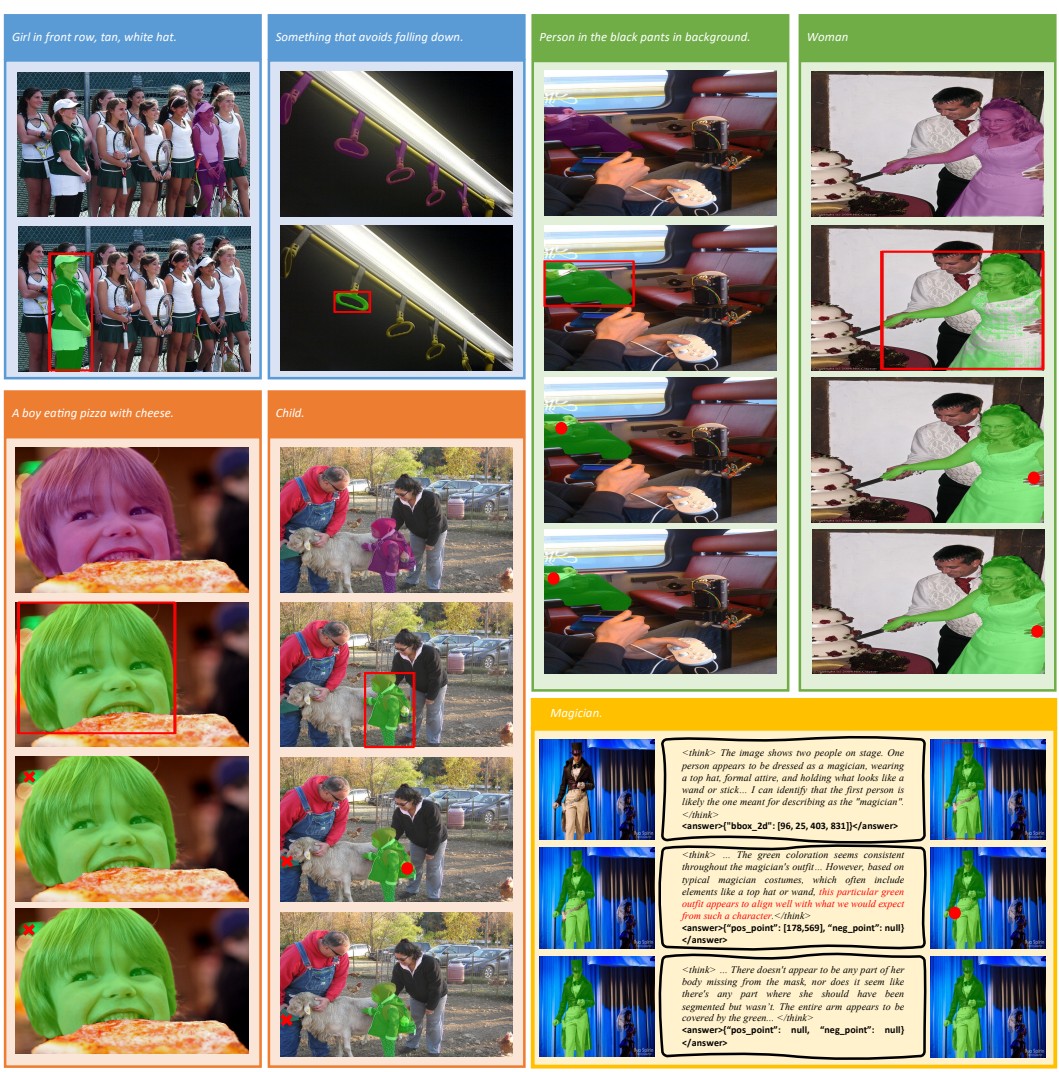

Figure 12: Failure cases of SAM-Veteran. Different types of errors are highlighted in different colors. The GT mask is shown in purple.

