# OpenReview forum: "SAM-Veteran: An MLLM-Based Human-like SAM Agent for Reasoning Segmentation"
_ICLR.cc/2026/Conference — ICLR 2026 Poster_

### Official Review · Reviewer_fRUK · 2025-10-22

**Soundness:** 3
**Presentation:** 4
**Contribution:** 3
**Rating:** 8
**Confidence:** 5

**Summary:**

This paper introduces SAM-Veteran, a reasoning-driven segmentation framework that enables human-like interaction with the Segment Anything Model (SAM). The method integrates a multi-modal large language model (MLLM) with SAM to perform iterative segmentation through three key steps: generating bounding boxes from image–query pairs, proposing refinement points based on SAM-generated masks, and adaptively terminating the process. To train the SAM-Veteran, diverse reward functions are utilized. Extensive experiments demonstrate that SAM-Veteran achieves state-of-the-art performance across both in-domain and out-of-domain datasets, effectively leveraging SAM’s interactive segmentation strengths.

**Strengths:**

- The paper presents the first unified framework that integrates bounding box generation, iterative mask refinement, and adaptive termination into a single reasoning-driven segmentation process.
- It introduces a well-designed multi-task reinforcement learning framework to effectively train the SAM-Veteran workflow, enhancing both textual grounding and mask comprehension.
- The proposed method achieves state-of-the-art performance on both in-domain and out-of-domain datasets, demonstrating strong generalization and robustness.

**Weaknesses:**

- Recently, in the reasoning segmentation task, several datasets beyond ReasonSeg have been proposed. (MUSE [1] for multi-target cases and MMR [2] for part-level reasoning). It would be interesting to see whether SAM-Veteran demonstrates good generalization performance across these diverse reasoning segmentation scenarios.

[1] Ren, Zhongwei, et al. "Pixellm: Pixel reasoning with large multimodal model." CVPR  2024.

[2] Jang, Donggon, et al. "Mmr: A large-scale benchmark dataset for multi-target and multi-granularity reasoning segmentation." ICLR 2025.

**Questions:**

Please refer to Weaknesses.

---

> ### Author Response · Authors · 2025-11-20
> **Response to Reviewer fRUK**
>
> We thank the reviewer for the valuable suggestion. We have evaluated SAM-Veteran on the recommended MMR and MUSE benchmarks.
>
> For MMR, we adopt the object-only version for fair comparison. Because an MMR query may refer to multiple objects, we merge the corresponding instance masks and treat the merged mask as the ground-truth segmentation.
>
> For MUSE, its original setting involves multi-target and multi-referring segmentation, which differs from the widely-used  benchmarks. To align with our evaluation protocols, we convert each sample into a classic triplet of image, single-object mask, and mask caption. We then feed the image and mask caption into the model and compare the predicted mask with the single-object ground-truth mask.
>
> The following table shows the results of different models on the two benchmarks.  Qwen+SAM2 provides the baseline performance on both datasets. Seg-Zero delivers a substantial improvement over this baseline. In contrast, SegAgent performs poorly on MUSE and nearly fails on MMR, reflecting the limited generalization ability introduced by its SFT-based training. Our SAM-Veteran further surpasses Seg-Zero by a clear margin on both benchmarks, demonstrating the effectiveness of our RL-based multi-task training framework.
>
> | Model       | MMR-object only |       | MUSE-test set |       |
> |-------------|-----------------|-------|---------------|-------|
> |             | gIoU            | cIoU  | gIoU          | cIoU  |
> | Qwen+SAM2   | 33.18           | 26.63 | 43.28         | 45.79 |
> | Seg-Zero    | 37.91           | 29.44 | 52.16         | 54.38 |
> | SegAgent    | 13.34           | 16.56 | 37.29         | 43.48 |
> | SAM-Veteran | 40.38           | 30.74 | 53.63         | 57.42 |
>
> **We have included the comparisons in the revised version. Please refer to Appendix B.1 for details.**
>
> *We hope that our responses adequately address the reviewer’s concerns. If any further clarification is needed, we would be happy to provide additional details at any time.*

---

> > ### Comment · Reviewer_fRUK · 2025-11-26
> > **Response to authors**
> >
> > Thank you for addressing my concerns. I believe the paper is worthy of acceptance to ICLR.

---

### Official Review · Reviewer_yz1X · 2025-10-25

**Soundness:** 3
**Presentation:** 3
**Contribution:** 3
**Rating:** 6
**Confidence:** 4

**Summary:**

This paper proposes SAM-Veteran, an MLLM-based agent that mimics human interaction with Segment Anything Model, enabling a complete “box generation to iterative refinement to adaptive termination” reasoning segmentation workflow. Built on a multi-task RL framework using Group Relative Policy Optimization, the method enhances textual grounding and mask comprehension, achieving new state-of-the-art performance and strong cross-domain generalisation.

**Strengths:**

1. Comprehensive experiments. The authors conduct extensive evaluations across multiple in-domain and out-of-domain datasets, providing solid empirical evidence for their claims.

2. Clear visualisations. The workflow and results are illustrated with intuitive and well-structured figures, making it easy to understand the model’s behaviour.

3. Well-organised and easy to follow. The paper is logically structured, with each component of the method introduced and justified clearly.

4. Strong motivation and validation. The motivation is well grounded, and the proposed design is effectively validated through both quantitative results and qualitative analysis.

**Weaknesses:**

1. Missing related works. Some relevant literature, such as recent works on task-generic promptable segmentation[1][2] and Grounding SAM[3], is not discussed or compared in the related work section. This weakens the contextual positioning of the contribution.

2. Insufficient hyperparameter analysis. Many components involve manually set hyperparameters (e.g., the threshold of R_{iou}^B=
0.4), but the paper does not explain how these values were chosen or their sensitivity.

3. Potential training complexity and stability issues . The framework integrates multiple tasks simultaneously, which may lead to excessive complexity. The paper does not provide sufficient analysis on training stability or convergence behaviour.

[1] Hu, Jian, et al. "Relax image-specific prompt requirement in sam: A single generic prompt for segmenting camouflaged objects." Proceedings of the AAAI Conference on Artificial Intelligence. Vol. 38. No. 11. 2024.

[2] Tang, Lv, et al. "Chain of visual perception: Harnessing multimodal large language models for zero-shot camouflaged object detection." Proceedings of the 32nd ACM international conference on multimedia. 2024.

[3] Ren, Tianhe, et al. "Grounded sam: Assembling open-world models for diverse visual tasks." arXiv preprint arXiv:2401.14159 (2024).

**Questions:**

1. Training stability. Given the multi-task RL design, how stable is training in practice? Are there signs of task interference or convergence issues?

2. Generalisation & scalability. How well does the method scale with more refinement steps or larger models, and can it generalise to broader segmentation scenarios beyond the tested datasets?

3. Hyperparameters & ablation. Many hyperparameters are manually set, but their selection process and sensitivity are unclear. Could the authors provide ablation or sensitivity analyses to justify these choices?

---

> ### Author Response · Authors · 2025-11-20
> **Response to Reviewer yz1X: Missing related works**
>
> **We have supplemented relevant citations in the revised manuscript and clarified the connections between our work and the three preceding foundational studies.** Grounded SAM pioneered the "detector + SAM" assembly paradigm, enabling open-vocabulary segmentation by fusing Grounding DINO with SAM; GenSAM proposed cross-modal chain-of-thought prompting and progressive mask generation to convert generic text into image-specific prompts for camouflaged object detection, advancing unsupervised SAM adaptation to specialized tasks; MMCPF enhanced MLLMs’ perception of camouflaged scenes through visual perception chains to enable zero-shot detection, validating the value of prompt engineering for visual localization. Building on these, our SAM-Veteran extends prior paradigm to a closed-loop "MLLM reasoning + SAM iterative segmentation" framework, allowing MLLM-SAM interaction like human users to adaptively optimize the results iteratively based on mask comprehension.

---

> > ### Author Response · Authors · 2025-11-20
> > **Response to Reviewer yz1X: Training Complexity & Stability**
> >
> > Regarding training complexity, SAM-Veteran is trained on three tasks, with the data for each task detailed in Appendix A.4. Specifically, for the grounding task and the auxiliary mask comprehension task, we use 9,000 samples for each. For the mask generation task, we use a total of 5 × (600 + 700) = 6,500 samples. The full training process (one episode) takes approximately 30 hours on 8 GPUs with 96 GB memory each. The overall training cost is manageable. **We have added the details of training requirements in Section 4.1 of the revised paper.**
> >
> > Regarding training stability, we adopt several strategies to ensure a robust optimization process:
> > 1. Task-specific dynamic sampling is designed to provide diverse actions with varying rewards, making optimization more stable and effective.
> > 2. We employ global batch normalization from REINFORCE++, instead of the local standard deviation used in GRPO.
> > 3. We balance the data distribution within each task. For example, in the mask-comprehension task, the ratio between “good-enough” cases and “need-refinement” cases is kept relatively balanced.
> > 4. We balance the amount of training data across tasks, mitigating interference between them.
> > 5. Carefully designed rewards and tuned hyperparameters further contribute to stable training.
> >
> > To illustrate the training dynamics of SAM-Veteran, **we present the curves of key metrics—such as entropy loss and reward—in the supplementary material**. The monitored values show that the reward increases steadily throughout training, with no signs of instability or convergence issues.

---

> ### Author Response · Authors · 2025-11-20
> **Response to Reviewer yz1X: Generalisation & Scalability**
>
> To evaluate the scalability of our method, we scale the MLLM from 7B to 32B and conduct experiments on Qwen2.5-VL-32B using the same settings as the 7B model. To further test scalability in the refinement step, we increase the maximum refinement steps from 3 to 5 for the 32B model. The results of the 32B variant across different datasets are presented in the following table. As observed, the 32B version of SAM-Veteran achieves further improvements on most in-domain and out-of-domain datasets, confirming the scalability of our approach. **We have included the experiments on scalability in Appendix B.3.**
>
> | Model           | ReasonSeg val |      | ReasonSeg test |      | RefCOCO testA | RefCOCO+ testA | RefCOCOg test |
> |-----------------|---------------|------|----------------|------|---------------|----------------|---------------|
> |                 | gIoU          | cIoU | gIoU           | cIoU | cIoU          | cIoU           | cIoU          |
> | SAM-Veteran-7B  | 68.2          | 67.3 | 62.6           | 56.1 | 80.8          | 76.6           | 73.4          |
> | SAM-Veteran-32B | 72.3          | 70.0 | 62.9           | 58.2 | 80.4          | 77.4           | 73.4          |
>
> Regarding generalization ability of SAM-Veteran, we include two more benchmarks—MME and MUSE—for a more thorough evaluation. For MMR, we adopt the object-only version for fair comparison. Because an MMR query may refer to multiple objects, we merge the corresponding instance masks and treat the merged mask as the ground-truth segmentation. For MUSE, its original setting involves multi-target and multi-referring segmentation, which differs from the widely-used  benchmarks. To align with our evaluation protocols, we convert each sample into a classic triplet of image, single-object mask, and mask caption. We then feed the image and mask caption into the model and compare the predicted mask with the single-object ground-truth mask. The following table shows the results of different models on the two benchmarks. Qwen+SAM2 provides the baseline performance on both datasets. Seg-Zero delivers a substantial improvement over this baseline. In contrast, SegAgent performs poorly on MUSE and nearly fails on MMR, reflecting the limited generalization ability introduced by its SFT-based training. Our SAM-Veteran further surpasses Seg-Zero by a clear margin on both benchmarks, demonstrating the effectiveness of our RL-based multi-task training framework.
>
> | Model       | MMR-object only |       | MUSE-test set |       |
> |-------------|-----------------|-------|---------------|-------|
> |             | gIoU            | cIoU  | gIoU          | cIoU  |
> | Qwen+SAM2   | 33.18           | 26.63 | 43.28         | 45.79 |
> | Seg-Zero    | 37.91           | 29.44 | 52.16         | 54.38 |
> | SegAgent    | 13.34           | 16.56 | 37.29         | 43.48 |
> | SAM-Veteran | 40.38           | 30.74 | 53.63         | 57.42 |
>
> **We have included the comparisons in the revised version. Please refer to Appendix B.1 for details.**

---

> ### Author Response · Authors · 2025-11-20
> **Response to Reviewer yz1X: Hyperparameters & Ablation**
>
> We conduct more analysis on hyperparamters as follows.
>
> | Parameter                               | Value                                  | ReasonSeg val | ReasonSeg test | RefCOCO testA | RefCOCO+ testA | RefCOCOg test | Avg. |
> |-----------------------------------------|----------------------------------------|---------------|----------------|---------------|----------------|---------------|------|
> | Baseline                                |                                        | 68.2          | 62.6           | 80.8          | 76.6           | 73.4          | 72.3 |
> | Task 1 $R^{\mathrm{B}}\_{\mathrm{IoU}}$ | Hard 0.7                               | 65.7          | 61.2           | 80.5          | 75.9           | 73.0          | 71.3 |
> |                                         | Hard 0.3                               | 67.1          | 61.2           | 80.6          | 77.3           | 72.5          | 71.7 |
> |                                         | Soft                                   | 65.9          | 60.0           | 80.2          | 75.7           | 72.0          | 70.7 |
> | Task 2 Reward                           | $R^{\Delta} = 3\cdot\mathrm{ReLU}(\Delta)$ | 68.1          | 61.5           | 80.2          | 77.0           | 73.2          | 72.0 |
> |                                         | $R^{\mathrm{ENC}}=0$                  | 66.8          | 62.0           | 80.2          | 76.6           | 72.9          | 71.7 |
> | Task 3 $\tau_{d}$                      | 10                                     | 67.3          | 62.1           | 80.3          | 76.6           | 72.0          | 71.6 |
> |                                         | 30                                     | 68.3          | 61.8           | 79.9          | 76.4           | 72.8          | 71.8 |
> | Rollout                                 | 4                                      | 66.5          | 62.0           | 79.9          | 76.1           | 72.0          | 71.3 |
> |                                         | 16                                     | 66.4          | 61.9           | 80.3          | 76.7           | 73.4          | 71.7 |
>
> **Task 1 $R^{\mathrm{B}}\_{\mathrm{IoU}}$.** We explore different designs of $R^{\mathrm{B}}\_{\mathrm{IoU}}$ in Task 1. Specifically, we evaluate several hard-threshold settings—0.3, 0.7, and the baseline 0.5—as well as a soft variant defined as $R^{\mathrm{B}}\_{\mathrm{IoU}} = \mathrm{IoU}(b, b^{\mathrm{GT}})$. The results indicate that the soft formulation performs worse than all hard-threshold versions, and among the hard thresholds, 0.5 yields the best performance.
>
> **Task 2 Reward.** First, we replace $R^{\Delta}$ in Task 2 with a linear variant whose maximum reward is 3, increasing linearly from 0 with respect to the IoU change $\Delta$, i.e., $R^{\Delta} = 3\cdot\mathrm{ReLU}(\Delta)$. Second, we set the encouraging score to $R^{\mathrm{ENC}} = 0$. Both modifications lead to a slight decrease in performance compared to the baseline.
>
> **Task 3 $\tau_{d}$.** We experiment with different values of $\tau_{d}$ in Task 3—10, 30, and the baseline 50. The results show that $\tau_{d} = 50$ achieves the best overall performance.
>
> **Training Rollout.** For the training hyperparameters, we experiment with different numbers of rollouts in GRPO—4, 16, and the baseline 8. The results show that using 8 rollouts yields the best overall performance.
>
> **We have included the above analysis Appendix B.4.**

---

> > ### Author Response · Authors · 2025-11-20
> > **Response to Reviewer yz1X**
> >
> > *We hope that our responses adequately address the reviewer’s concerns, and we kindly invite the reviewer to reconsider and improve the rating of our paper. If any further clarification is needed, we would be happy to provide additional details at any time.*

---

> > > ### Comment · Reviewer_yz1X · 2025-11-27
> > > **Response to authors**
> > >
> > > Thanks for the author's response, I believe your detailed feedback has addressed most of my concerns, so I would like I maintain my positive score.

---

### Official Review · Reviewer_GtqV · 2025-10-26

**Soundness:** 3
**Presentation:** 3
**Contribution:** 3
**Rating:** 8
**Confidence:** 4

**Summary:**

This paper presents SAM-Veteran as an MLLM-based solution for reasoning segmentation, of which the key feature is to emulate the human SAM users to refine the mask iteratively. Instead of relying on supervised fine-tuning of the MLLM, SAM-Veteran is built on top of the current RL-based training paradigms. The paper proposes to divide the iterative reasoning segmentation task into three sub-tasks and design reward models to help train the MLLM on each of those sub-tasks. The proposed method shows strong results on several reasoning segmentation benchmarks compared to prior SFT and RL-based methods.

**Strengths:**

The paper is well-written, well-organized, and easy to follow. In general, the proposed method shows noticeable improvement over the prior RL-based methods and SFT-based methods, and is closer to actual human interaction with the SAM-like segmentation models. The paper includes a thorough experiment for analyzing the contribution of each design choice, quantitative results, and qualitative visualizations to clearly show the strength and potential failing cases of the proposed method.

**Weaknesses:**

The motivation for having an auxiliary task is clear; however, the improvement is not significant, as shown in Table 4. Meanwhile, this means that there is room for improvement in task 2. A better reward design or pipeline for mask comprehension is needed. This weakness, as the paper points out, is linked to the limited ability of MLLM to understand images with masks, and adding a specific type of color mask on the visual input directly will inevitably compromise the valuable information contained in the target region.

**Questions:**

Please see the weaknesses.

---

> ### Author Response · Authors · 2025-11-20
> **Response to Reviewer GtqV**
>
> We thank the reviewer for the valuable comment.
>
> Table 4 illustrates the effect of the auxiliary task on segmentation IoU and the model’s termination behavior. Although the auxiliary task yields only minor changes in IoU, it is crucial for enabling proper termination: without it, the model produces positive points for arbitrary inputs, whereas with it, the model learns to adaptively terminate when the mask is good enough. Therefore, the auxiliary task is essential to our training framework.
>
> For our mask-comprehension strategy, we follow the design of SegAgent and overlay a green mask onto the image. SegAgent also experiments with other overlay colors and reports that the choice of color has minimal impact. We agree with the reviewer that this approach is limited, as it alters the original color information—an issue explicitly acknowledged in our limitation section (Appendix C). In addition, green regions in the input image may be incorrectly interpreted as green masks, as illustrated in our failure case study (Appendix B.7).
>
> Beyond the current approach, we explored several alternative ways to present the mask to the MLLM, including: (1) highlighting the masked region, and (2) providing the mask as a separate image (either a binary mask or a masked image with a black background). We found that the first alternative achieves performance comparable to our current method. However, it also exhibits limitations, particularly when the question concerns illumination-related attributes—for example, queries such as “the lighter table lamp.” For the second alternative, the MLLM struggled to interpret the separate mask input and produced poor results. We believe this is because most base MLLMs (e.g., Qwen-2.5VL) are not trained for tasks that require jointly understanding two images—the original image and an accompanying mask image. Consequently, making this second alternative work would essentially require enhancing the base model’s fundamental capability, potentially through additional supervised fine-tuning (SFT), which risks catastrophic forgetting of general reasoning abilities and poor generalization to out-of-domain data.
>
> While we fully acknowledge this limitation, we plan to explore more sophisticated ways of integrating the mask into the input in future work, without compromising either the information in the original image or the capability of the MLLM. A potential direction is to combine SFT with RL and introduce a dedicated mask token, enabling the model to acquire the fundamental ability to understand image masks. We sincerely appreciate the reviewer for highlighting this structural limitation, which provides a valuable and inspiring direction for future research.
>
> *We hope that our responses adequately address the reviewer’s concerns. If any further clarification is needed, we would be happy to provide additional details at any time.*

---

### Official Review · Reviewer_yXBo · 2025-10-30

**Soundness:** 3
**Presentation:** 3
**Contribution:** 3
**Rating:** 6
**Confidence:** 4

**Summary:**

This paper introduces SAM-Veteran, a multi-modal large language model (MLLM)-based agent designed to emulate human-like interaction with the Segment Anything Model (SAM) for reasoning segmentation. The key contributions include:

- A **multi-task reinforcement learning (RL) framework** that trains the MLLM to generate bounding boxes, refine masks iteratively via points, and adaptively terminate the process.
- A **dynamic sampling strategy** to stabilize training by diversifying box/point generation.
- State-of-the-art performance on in-domain (RefCOCO) and out-of-domain (ReasonSeg) benchmarks, demonstrating improved generalization over existing supervised fine-tuning (SFT) and RL-based methods.

**Strengths:**

**Originality**: The work creatively combines MLLMs with SAM’s interactive segmentation capabilities, addressing a gap in prior RL-based methods that neglect iterative refinement. The integration of three RL tasks (grounding, mask comprehension, and auxiliary) to mimic human-SAM interaction is novel.
**Quality**: The experiments are thorough, including ablation studies on reward design, multi-task training, and dynamic sampling. The comparison with SFT and RL baselines (e.g., Seg-Zero, SAM-R1) validates the framework’s effectiveness.
**Clarity**: The paper is well-structured, with clear task formulations (e.g., MDP definitions) and visualizations of the workflow. The appendices provide implementation details, prompts, and failure case analysis.
**Significance**: The work advances MLLM-based segmentation by enabling human-like SAM usage, which could inspire future research on interactive vision-language systems. The code and configuration details (Figure 6) enhance reproducibility.

**Weaknesses:**

**Limited comparison with SegAgent**: While SegAgent (Zhu et al., 2025b) is mentioned, the paper does not quantitatively compare SAM-Veteran against it, despite both addressing iterative refinement. This omission weakens the claim of novelty.

**Inference time:** The paper does not provide inference time comparisons with baseline methods, which is critical for evaluating practical deployment feasibility. Additionally, detailed training resource requirements (e.g., GPU hours, memory consumption) are not explicitly reported, limiting reproducibility assessments and computational cost analysis for researchers with constrained resources.

**SAM dependency**: The framework heavily relies on SAM for mask generation and reward computation. The impact of SAM’s inherent limitations (e.g., failure modes on fine-grained objects) on SAM-Veteran’s performance is not discussed.
**User study absence**: While the paper emphasizes "human-like" behavior, no user study evaluates whether the termination policy or refinement steps align with human preferences.

**Questions:**

**Q1**: How does SAM-Veteran compare to SegAgent in terms of refinement steps and termination accuracy? The authors should include a direct comparison to clarify their method’s advantages.
**Q2**: Could the inference time and training requirement be provided? This would potentially enhance the feasibility of real-world deployment.

**Q3**: The paper states that SAM-Veteran avoids "catastrophic forgetting" seen in SFT methods. Is there empirical evidence (e.g., performance on general MLLM benchmarks) to support this claim?

---

> ### Author Response · Authors · 2025-11-20
> **Response to Reviewer yXBo: Comparisons to SegAgent**
>
> We conduct comprehensive comparisons between SAM-Veteran and SegAgent.
>
> For segmentation metrics, because SegAgent does not report out-of-domain results (ReasonSeg val and ReasonSeg test), we evaluate the officially released SegAgent checkpoint and include its performance in the table below. As expected, it performs poorly on these out-of-domain benchmarks, likely due to its reliance on STAR+, an RL-style method that is essentially based on SFT for optimizing the MLLM. **We have included the results in Table 3 and the discussion in the revised paper.**
>
> In addition, we include two more benchmarks—MME and MUSE—for a more thorough evaluation. For MMR, we adopt the object-only version for fair comparison. Because a query in MMR may refer to multiple objects, we merge the corresponding instance masks and treat the merged mask as the ground-truth segmentation. For MUSE, its original setting involves multi-target and multi-referring segmentation, which differs from the widely used  benchmarks. To align with our evaluation protocols, we convert each sample into a classic triplet of image, single-object mask, and mask caption. We then feed the image and mask caption into the model and compare the predicted mask with the single-object ground-truth mask. The table below reports the results on these two benchmarks. As shown, SAM-Veteran substantially outperforms SegAgent on both datasets, further validating the effectiveness of our RL-based multi-task training framework. **We have included the comparisons in the revised version. Please refer to Appendix B.1 for details.**
>
> | Model       | ReasonSeg val |       | ReasonSeg test |      | MMR-object only |       | MUSE-test set |       |
> |-------------|---------------|-------|----------------|------|-----------------|-------|---------------|-------|
> |             | gIoU          | cIOU  | gIoU           | cIOU | gIoU            | cIOU  | gIoU          | cIOU  |
> | Seg-Agent   | 33.0          | 25.4  | 33.5           | 31.3 | 13.34           | 16.56 | 37.29         | 43.48 |
> | SAM-Veteran | 68.2          | 67.3  | 62.6           | 56.1 | 40.38           | 30.74 | 53.63         | 57.42 |
>
> For qualitative comparisons, we visualize the mask prediction process of SegAgent and SAM-Veteran. **Please refer to the visualization in Appendix B.6 (Figure 11).** As illustrated, SegAgent performs a fixed seven refinement steps and sometimes generates ineffective or irrelevant points. In contrast, SAM-Veteran consistently generates accurate bounding boxes and refinement points, while also being able to dynamically determine when to terminate the procedure. These qualitative results further highlight the advantages of our method over SegAgent.
>
> Regarding the comparison of refinement step and inference time consumption, please refer to the next response.

---

> ### Author Response · Authors · 2025-11-20
> **Response to Reviewer yXBo: Training and inference complexity**
>
> We compare the inference-time efficiency of different models using two metrics: the average number of MLLM inference steps and the average time cost per sample on the RefCOCO testA dataset. The results are shown in the following table. For Qwen+SAM, we report the results of generating boxes for SAM. For Seg-Zero, the MLLM outputs both the bounding boxes and the points for SAM in a single step, whereas SegAgent adopts a fixed number of 7 refinement iterations for mask prediction. As shown in the results, Qwen+SAM and Seg-Zero finish the task in a single step (\~3.5s), but their segmentation performance is inferior to that of multi-step refinement methods, as evidenced in Table 3. SegAgent, on the other hand, requires a fixed 7-step MLLM inference pipeline (~9s), leading to low efficiency. Our SAM-Veteran achieves stronger performance with substantially fewer steps (<2.5 steps on average, ~5s). Our method is slower per sample per step than SegAgent because CoT introduces more response tokens, resulting in improved performance at the cost of extra time. Nevertheless, our method strikes a more favorable balance between segmentation quality and inference efficiency. Furthermore, we improve efficiency by replacing the Transformer backend with the vLLM backend, which significantly reduces the inference time—approximately cutting the time consumption in half. This optimization is applied to all evaluations in our experiments. **We have included the discussion on inference complexity in Appendix B.5 of the revised paper.**
>
> | Method      | Inference backend | Avg. Step | Avg. Time (s) |
> |-------------|-------------------|-----------|---------------|
> | Qwen+SAM2   | Transformers      | 1         | 3.11          |
> | SegZero     | Transformers      | 1         | 3.43          |
> | SegAgent    | Transformers      | 7         | 8.95          |
> | SAM-Veteran | Transformers      | 2.08      | 5.09          |
> | SAM-Veteran | vLLM              | 2.21      | 2.47          |
>
> Regarding the training requirements, SAM-Veteran is trained on three tasks, with the data for each task detailed in Appendix A.4. Specifically, for the grounding task and the auxiliary mask comprehension task, we use 9,000 samples for each. For the mask generation task, we use a total of 5 × (600 + 700) = 6,500 samples. The full training process (one episode) takes approximately 30 hours on 8 GPUs with 96 GB memory each. **We have added the details of training requirements in Section 4.1 of the revised paper.**

---

> > ### Author Response · Authors · 2025-11-20
> > **Response to Reviewer yXBo: The claim of "avoid catastrophic forgetting"**
> >
> > To substantiate our claim that the proposed RL framework mitigates catastrophic forgetting of general reasoning ability, we evaluate SAM-Veteran—along with several baseline models—on standard general-purpose MLLM benchmarks (results shown in the table below, * means re-evaluation in our environment). As observed, the RL-based methods, Seg-Zero and our SAM-Veteran, maintain performance comparable to their respective base MLLM (Qwen2.5-VL), demonstrating that RL training preserves general reasoning capability. In contrast, the SFT-based model SegAgent exhibits a clear decline in performance on general vision-language benchmarks relative to its base model (Qwen-VL-Chat), indicating significant loss of generalization ability. These results confirm that SFT-based training is prone to catastrophic forgetting, whereas RL-based methods effectively avoid this issue. **We have included the experiment and discussion in Appendix B.2 of the revised paper.**
> >
> > |                                   | Qwen2.5 | Qwen2.5* | Seg-Zero* | SAM-Veteran | Qwen-VL-Chat* | SegAgent* |
> > |-----------------------------------|---------|----------|-----------|-------------|---------------|-----------|
> > | OCR-related Understanding Tasks   |         |          |           |             |               |           |
> > | SEEDBench2_Plus                   | 70.4    | 69.6     | 69.5      | 69.5        | 44.6          | 9.7       |
> > | TextVQA_VAL                       | 84.9    | 85.3     | 85.4      | 84.2        | 60.2          | 1.29      |
> > | General Visual Question Answering |         |          |           |             |               |           |
> > | MMStar                            | 63.9    | 59.9     | 61.3      | 60.5        | 29.0          | 5.3       |
> > | MME                               | 2347    | 2303     | 2286      | 2328        | 1834          | 753       |
> > | MUIRBench                         | 59.6    | 58.3     | 57.4      | 59.2        | 27.9          | 12.23     |
> > | BLINK                             | 56.4    | 54.3     | 55.3      | 54.0        | 14.4          | 4.42      |
> > | RealWorldQA                       | 68.5    | 67.8     | 68.9      | 66.0        | 45.8          | 1.57      |

---

> ### Author Response · Authors · 2025-11-20
> **Response to Reviewer yXBo**
>
> *We hope that our responses adequately address the reviewer’s concerns, and we kindly invite the reviewer to reconsider and improve the rating of our paper. If any further clarification is needed, we would be happy to provide additional details at any time.*

---

### Author Response · Authors · 2025-11-21
**Response to reviewers**

We sincerely thank all reviewers for their valuable comments and suggestions.
We have carefully addressed each concern point by point, and the corresponding revisions have been incorporated into the updated manuscript. In addition, we provide further visualizations of our method in the supplementary materials.

We hope that our responses satisfactorily resolve the reviewers’ concerns. If any additional clarification is needed, we would be glad to provide further details.

---

### Author Response · Authors · 2025-12-01
**Response to Area Chair**

We sincerely thank you for overseeing the review process of our submission and for the reviewers’ careful and constructive feedback. We would like to briefly summarize the review outcomes and how the raised concerns were addressed in our rebuttal.

Overall, the submission received consistently positive evaluations. All reviewers rated the paper as technically sound, clearly presented, and of meaningful contribution. Two reviewers recommended acceptance with high confidence (8/10), and the remaining reviewers provided borderline-accept scores (6/10), which were later maintained or strengthened after the rebuttal.

The main concerns focused on (1) comparisons with SegAgent, (2) inference efficiency and training cost, (3) empirical evidence for avoiding catastrophic forgetting, (4) training stability, scalability, and hyperparameter sensitivity, and (5) evaluation on additional datasets and related work coverage. We addressed each of these points with substantial additions and clarifications, summarized below.
1. We conducted comprehensive quantitative and qualitative comparisons with SegAgent using its official checkpoint. Results on ReasonSeg (val/test), MMR (object-only), and MUSE demonstrate that SAM-Veteran significantly outperforms SegAgent, particularly in out-of-domain scenarios. Visual comparisons further show that SAM-Veteran generates more reliable refinement points and learns adaptive termination, while SegAgent relies on fixed and often inefficient refinement steps.
2. We reported detailed inference-time and training-cost analyses. SAM-Veteran requires fewer than 2.5 inference steps on average, compared to SegAgent’s fixed 7-step pipeline, achieving a better balance between efficiency and segmentation quality. We also included absolute time measurements and full training resource requirements.
3. To substantiate our claim that RL mitigates catastrophic forgetting, we evaluated SAM-Veteran and baseline models on standard general-purpose MLLM benchmarks (e.g., SEEDBench2+, TextVQA, MME, MMStar). The results show that RL-based methods preserve general reasoning abilities comparable to their base models, whereas the SFT-based SegAgent suffers severe performance degradation, confirming the claim empirically.
4. In response to concerns about training complexity and stability, we detailed the multi-task training setup, stabilization strategies (dynamic sampling, task balancing, global normalization), and provided training curves showing stable convergence. We also conducted extensive hyperparameter and ablation studies to justify key design choices and thresholds.
5. We expanded evaluations to additional reasoning segmentation benchmarks (MMR and MUSE), demonstrated scalability from 7B to 32B models, and supplemented the related work section to better contextualize our contribution relative to recent studies.

Following these responses, reviewers explicitly acknowledged that their concerns had been addressed, with one reviewer stating that the paper is worthy of acceptance to ICLR, and another maintaining a positive score after the rebuttal.

We hope this summary is helpful for your final assessment. We sincerely appreciate your time and consideration.

---

### Public Comment · ~Jie_Liu21 · 2026-02-27
**Request for code release**

Dear authors,

Congratulations on your ICLR2026 paper!

This work is truly inspiring, do you have plan to release the code to facilitate future research?

Cheers,
Jie

---

### Meta-Review · Area_Chair_SG45 · 2026-01-06

**Summary:**

This paper proposes SAM-Veteran, a multi-modal large language model  (MLLM)-based agent that aims to emulate human-like interaction with the Segment Anything Model (SAM). The proposed method performs iterative segmentation through three steps: (1) generating bounding boxes from an image–query pair, (2) iteratively proposing refinement points based on SAM-produced masks, and (3) adaptively terminating the process. The method is trained with a multi-task RL framework based on GRPO, and introduces dynamic sampling for box/point generation to stabilize optimization. Experiments indicate strong performance on both in-domain and out-of-domain benchmarks.

**Reviewer Concerns:**

This paper has four positive initial recommendations, including two borderlines and two accepts. The reviewers generally find the submission technically sound, clearly written, and empirically thorough.

There were several concerns raised by the reviewers and most of them seems to be addressed by the rebuttal comment.

- Comparison with SegAgent (yXBo)
- Training and inference complexity (yXBo, yz1X)
- Lack of evidence of the "catastrophic forgetting" claim (yXBo)
- Additional benchmarks beyond ReasonSeg (MUSE and MMR) (fRUK)

The rebuttal comment provided additional empirical results to address the concerns. In my opinion, most of the significant concerns raised by the reviewers are resolved by the rebuttal comment.

Overall, I think the advantages of this paper (novel integrated closed-loop formulation and strong empirical results) outweigh its disadvantages (dependency on SAM and MLLM, the "human-like" argument is not well supported). Hence, I recommend acceptance for this paper.

**Reviewer Scores:**

I guess that if the reviewers would have changed their score above the borderline accept if they had been able to participate fully in the discussion. Specifically, after the rebuttal, at least one borderline reviewer explicitly noted that most concerns were addressed and maintained a positive score, and another reviewer stated the paper is worthy of acceptance.

---

### Decision · Program_Chairs · 2026-01-26

Accept (Poster)